# Optical properties of coated black carbon aggregates: numerical simulations, radiative forcing estimates, and size-resolved parametrization scheme

Baseerat Romshoo[1], Thomas Müller[1], Sascha Pfeifer[1], Jorge Saturno[2], Andreas Nowak[2], Krzysztof Ciupek[3], Paul Quincey[3], and Alfred Wiedensohler[1]

[1]Leibniz Institute for Tropospheric Research, 04318, Leipzig, Germany

[2]PTB Physikalisch-Technische Bundesanstalt, 38116, Braunschweig, Germany

[3]Environment Department, National Physical Laboratory (NPL), Teddington, TW11 0LW, UK

*Correspondence to*: Baseerat Romshoo (baseerat@tropos.de)

**Abstract.** The formation of black carbon fractal aggregates (BCFAs) from combustion and subsequent aging involves several stages resulting in modifications of particle size, morphology, and composition over time. To understand and quantify how each of these modifications influences the BC radiative forcing, the optical properties of BCFAs are modelled. Owing to the high computational time involved in numerical modelling, there are some gaps in terms of data coverage and knowledge regarding how optical properties of coated BCFAs vary over the range of different factors (size, shape, and composition). This investigation bridged those gaps by following a state-of-the-art description scheme of BCFAs based on morphology, composition, and wavelength. The BCFAs optical properties were investigated as a function of the radius of the primary particle ($a_o$), fractal dimension ($D_f$), fraction of organics ($f_{organics}$), wavelength ($\lambda$), and mobility diameter ($D_{mob}$). The optical properties are calculated using the multiple sphere T-matrix (MSTM) method. For the first time, the modelled optical properties of BC are expressed in terms of mobility diameter ($D_{mob}$), making the results more relevant and relatable for ambient and laboratory BC studies. Amongst size, morphology, and composition, all the optical properties showed the highest variability with changing size. The cross-sections varied from 0.0001 $\mu m^2$ to 0.1 $\mu m^2$ for BCFA $D_{mob}$ ranging from 24 nm to 810 nm. It has been shown that $MAC_{BC}$ and $SSA$ is sensitive to morphology especially for larger particles with $D_{mob} > 100$nm. Therefore, while using the simplified core-shell representation of BC in global models, the influence of morphology on radiative forcing estimations might not be adequately considered. The Ångström absorption exponent varied from 1.06 up to 3.6 and increased with the fraction of organics ($f_{organics}$). Measurement results of AAE >> 1 are often misinterpreted as biomass burning aerosol, it was observed that the AAE of purely black carbon particle can be >> 1 in the case of larger BC particles. The values of the absorption enhancement factor ($E_\lambda$) via coating were found between 1.01 and 3.28 in the visible spectrum. The $E_\lambda$ was derived from Mie calculations for coated volume equivalent spheres, and from MSTM for coated BCFAs. Mie calculated enhancement factors were found to be larger by a factor of 1.1 to 1.5 than their corresponding values calculated from the MSTM method. It is shown that radiative forcings are highly sensitive towards modifications in morphology and composition. The black carbon radiative forcing $\Delta F_{TOA}$ ($Wm^{-2}$) decreases up to 61% as the BCFA becomes more compact, indicating that the global model calculations should account for changes in morphology. A decrease of more than 50% in $\Delta F_{TOA}$ was observed as the organic content of the particle increase up to 90%. The changes in the ageing factors (composition and morphology) in tandem result in an overall decrease in the $\Delta F_{TOA}$. A parameterization scheme for optical properties of BC fractal aggregates was developed, which is applicable for modelling, ambient and laboratory-based BC studies. The parameterization scheme for the cross-sections (extinction, absorption, and scattering), single scattering albedo ($SSA$), and asymmetry parameter ($g$) of pure and coated BCFAs as a function of $D_{mob}$ were derived from tabulated results of the MSTM method. Spanning over an extensive parameter space, the developed parametrization scheme showed promisingly high accuracy up to 98% for the cross-sections, 97% for single scattering albedos ($SSA$), and 82% for asymmetry parameter ($g$).

## 1. Introduction

Black carbon (BC), also called light-absorbing carbon (LAC), is produced from incomplete combustion of fossil fuels, biomass, and biofuels, and is reported to be the second largest contributor to global warming after $CO_2$ with the global forcing estimates ranging between 0.4 to 1.2 $W/m^2$ (Ramanathan and Carmichael, 2008). It has been found that the annual anthropogenic BC emissions have increased from 6.6 to 7.2 tera-grams during 2000-2010 (Klimont et al., 2017). Moreover, due to rapid urbanization in many developing regions like China, South Asia, South East Asia, the total aerosol mas constitutes of a significantly large portion of BC (Kumar et al., 2018; Bond

et al., 2007; Wiedensohler et al., 2002; Madueno et al., 2019, 2020). In addition to the warming effect, BC also
decreases snow albedo (Doherty et al, 2010), causes adverse health effects (Janssen et al., 2011), and lowers
visibility (Wang et al., 2020).
Optical properties of BC are of scientific interest because they allow conclusions to be drawn on the nature of
the particles and to investigate their radiative impacts (Liu et al., 2015; Safai et al., 2015). After its emission into
the atmosphere, BC particles undergo various changes in shape, size, and composition (Fierce et al., 2013). In the
early stages of formation, BC particles consist of loosely bound agglomerates made of numerous small spherules,
which collide to form strongly bound chain-like aggregates (Michelsen et al., 2017). Depending upon the
atmospheric conditions after emission, irregularly shaped primary spherules provide active sites for the
deposition of water vapour which causes changes in the hygroscopicity of the particles (Petzold et al., 2005; Peng
et al., 2017, ). In addition to this, different by-products of combustion like organic vapours are deposited around
the particles (Siegmann et al., 2002; Rudich et al., 2007). These processes lead to the formation of coatings on
BC cores (Bond et al., 2006) and reshaping of the BC particles into more spherical structures (Abel et al., 2003).
With the BC particles becoming more compact, an increase in the extinction cross section is observed (Liu et al.,
2012). It was theoretically shown in clusters of absorbing spherules that the change in the optical cross-sections
with an increasing number of spherules (aggregation) is strongly dependent on the morphology (Berry and
Percival, 1986). Laboratory and ambient studies also show changes in the optical properties of BC with an
increasing volume of organic coating (Shiraiwa et al., 2010; Cheng et al., 2009). Even though the organic coating
is less absorbing by nature, but an increase in the absorption cross section is observed due to the lensing effect
(Zhang et al., 2018; Zanatta et al., 2016, Saleh et al., 2015). Additionally, there exists a class of organic carbon
(OC) with light absorbing properties, known as brown carbon, strongly absorbing solar radiation in the blue and
near-ultraviolet spectrum (Fleming et al., 2020; Feng et al., 2004; Chakrabarty et al., 2010; Chen and Bond, 2010).
Numerical modelling has been proven to be helpful in better understanding the effect of the changes that BC
particles undergo on their optical properties (Scarnato et al., 2013; Kahnert, 2010; Smith and Grainger, 2014).
The advantage of the modelling studies is the ability and flexibility they offer to simulate BC particles of desired
size, shape, and composition, hence improving our understanding of BCFAs at the micro-physical level.
The representation of the simulated BC particle plays an essential role in their numerically derived optical
properties. The assumption of BC particles as spheres is widely used by atmospheric scientists, especially in the
field of climate modelling (Stier et al., 2004; Ma et al., 2011; Düsing et al.,2018;). In the case of aged BC, it is
commonly considered that a spherical BC core is encapsulated inside another sphere representing the coating.
This morphology is used in the core-shell Mie theory (Bohren and Huffman, 1983) for obtaining the optical
properties of such particles. Even though this method is simple, it might result in larger discrepancies when
compared to the actual measurements (Wu et al., 2018). Mie theory also overestimates absorption for core-shell
configuration of BC particles in the visible range of light (Adachi et al., 2010). It was shown that the ratio of non-
BC to BC components plays an important role in determining the performance of different methods used for
simulating the BC optical properties (Liu et al., 2017). Electron microscopy results of the samples from laboratory
and ambient measurements of BC (Ouf et al., 2016; Dong et al., 2018) showed that the BC particles consist of
agglomerates made up of numerous primary particles. It has been observed that these particles show self-similarity
when viewed over a range of scales, which is an important characteristic of fractals (Forrest and Witten, 1979).
This makes BC particles suitable to be termed as black carbon fractal aggregates (BCFAs), and is used as such
throughout this study.
Discrepancies due to Mie theory have caused an increasing interest in the simulation of the BC optical
properties assuming a more realistic fractal morphology. A size-dependent empirical formula for the optical
properties of BCFAs was derived for the wavelength range from 200nm up to 12.2μm (Kahnert et al., 2010). The
optical properties of pure BCFAs, i.e., without any coating, were investigated by Smith and Grainger (2014),
further developing a parametrization for optical properties of pure BCFAs with respect to the number of primary
particles ($N_s$). A method to estimate the optical properties BCFAs was proposed using the machine learning model
'support vector machine' (Luo et al., 2018). Empirical equations on the BC Ångström absorption exponent (AAE)
were derived for different BC morphologies (Liu et al., 2018). A database containing optical data was developed
that includes the aggregation structure, refractive index, and particle size of BCFAs (Liu et al., 2019).
Various ambient and laboratory studies have emphasized the role of organic external coating in influencing the
BC absorption and scattering properties (Zhang et al., 2008, Ouf et al., 2016; Dong et al., 2018, Shiraiwa et al.,
2010). However, the previous modelling-based studies were not able to take into account the information about
the coating of the BCFAs. The reason for this could be that the time-consuming simulations make the
computational load for such a task substantially large. It was also pointed out that improved size-resolved datasets
and models for the light absorbing carbon (LAC) is required that includes observables like optical properties,
OC/BC ratio, burning phase or fuel types (Liu et al., 2020). Therefore, a size-resolved parametrization scheme for
optical properties of BCFAs including the external coating parameter is very important.
This investigation involved computationally intensive modeling aimed at understanding and quantifying the
changes that BCFAs and their optical properties undergo by simulating various cases of the BCFAs under an

elaborated systematic approach that is designed to span a wide parameter space. The coating parameter is quantified through the fraction of organics ($f_{organics}$). The BCFA cases are classified according to various $f_{organics}$, morphologies, and wavelengths. This approach of categorization involving $f_{organics}$ of BCFAs is aimed to bridge the gaps that are present in the modeled optical data from the previous studies. The optical properties were calculated using the T-matrix code (Mackowski et al., 2013) and the findings are presented and discussed with respect to the equivalent mobility diameter ($D_{mob}$) making it more relevant and comparable for laboratory, and ambient studies in which mobility spectrometers are often used for size classification.

The study highlights how modifications in the morphology and $f_{organics}$ of BCFAs can further influence the BC radiative forcing. Finally, the parameterization scheme for optical properties (extinction, scattering, and absorption) of coated BCFAs was developed as a function of size for different morphologies, $f_{organics}$, and wavelengths.

## 2. Methods

### 2.1 Morphology of BCFAs

The formation of BCFAs from combustion is a process involving several stages. Along with BC, a complex mixture of gas-phase organic compounds with a spectrum of molecular structures are co-emitted during incomplete combustion (Siegmann et al., 2002; Gentner et al., 2017). Depending upon the source of burning, different types of polycyclic aromatic hydrocarbons (PAHs) are considered to be the direct pre-cursors of BCFAs (Bockhorn 2009). Small PAHs such as acetylene ($C_2H_2$) are attached to larger precursor PAHs resulting in the growth of these elementary structures. It is postulated that the nucleation of two large PAHs leads to the formation of small three-dimensional particles with diameters ranging from 1-2 nm (Calcote, 1981).

Processes like surface growth and coagulation of gaseous phase molecules or PAHs leads to further growth of these particles. High-resolution transmission electron microscopy (TEM) images revealed these particles to be spherules up to the diameter of 10-30nm specific to the flame (Homann, 1967). These primary particles show a randomly ordered microstructure of graphite layers (Hess et al., 1969). Following the processes of nucleation and coagulation, the primary particles form larger BCFAs, which subsequently grow by aggregation (Sorensen, 2001). Following this concept of fractal morphology, a mathematical description of fractal aggregates was formulated (Mishchenko et al., 2002) by:

$$N_s = k_f \left(\frac{R_g}{a_0}\right)^{D_f},$$ (1)

where, $a_o$ is the radius of primary particles, $N_s$ is the number of primary particles, $D_f$ is the fractal dimension, and $k_f$ is a fractal pre-factor. $R_g$ is the radius of gyration, which characterizes the spatial size of the aggregate. It is defined as root means square (rms) distance of the aggregate from its geometrical center by:

$$R_g^2 = \frac{1}{N_s} \sum_{i=1}^{N_s}(r_i - r_o)^2 \quad ,$$ (2)

where, $r_i$ is the position vector of the $i^{th}$ primary particle, and $r_o$ is the position vector of the center of mass of an aggregate with radius of gyration $R_g$.

The size of a BCFA is determined by two parameters, the radius of the primary particle ($a_o$) and number of primary particles ($N_s$). Both are sensitive to the emission source. BCFAs originating from the combustion of biomass have a radius of the primary particle varying between 15- 25 nm (Chakrabarty et al., 2006). On the other hand, emissions from aircraft turbines comprise of primary particles with a radius of 5 nm (Liati et al., 2014). Aggregates emitted from diesel engines have a radius of the primary particle varying between 10 nm and 12 nm (Guarieiro et al., 2018). Some experimental studies indicate that in the atmosphere, the radius of the primary particle is polydisperse in nature varying from 10-100nm (Bescond et al. 2014). Following these studies, Liu et al., 2015 reported differences in the optical properties of BCFAs due to the monodisperse and polydisperse distribution of the radii of the primary particles. Contrarily, Berry and Percival (1986) showed that light absorption measurements are insensitive to the radii of the primary particles. Additionally, Kahnert (2012b) pointed out that insensitivity is present when the radii of the primary particle fall in the range of 10 – 25nm. For the sake of simplicity, aggregates of monodisperse primary particle size were used in this study.

Further, the reshaping of BCFAs into collapsed, sphere-like structures while ageing can be described by the fractal dimension ($D_f$) (Sorensen, 2001). The value of $D_f$ increases as an aggregate reshapes into a more spherical particle. A $D_f$ of 3 being the value for a sphere, whereas $D_f$ of 1 represents an open-chain like aggregate. In the early stages of their formation, BCFAs have a fractal dimension ($D_f$) between 1.5 and 1.9 (China et al., 2014; Wentzel et al., 2003). However, as a consequence of the atmospheric aging, the aggregates transform from being

bare to partly coated, embedded in coatings. In this case, the fractal dimension can go up to 2.2 (Wang et al., 2017). The exposure to humidity and coatings can collapse the BCFA into a structure having even a larger fractal dimension up to 2.6. (Zhang et al., 2008; Bambha et al., 2013). Hence, studying BC particles under the assumption of aggregate morphology provides a wider range of parameter space (particle size, primary particle size, and morphology). This is limited to only particle size in case of spherical assumptions.

Aggregates are formed from the random motion of a cluster meeting cluster (Sorensen 2001). If the probability of sticking is considered 1, the process of formation is called the diffusion-limited cluster aggregation (Witten and Sander, 1983). Following this principle, Diffusion-limited algorithms (DLAs) have been developed, which include cluster-cluster aggregation (CCA) (Thouy and Julien, 1994) and particle-cluster aggregation (PCA) methods (Hentschel,1984). In this study, the tunable diffusion limited aggregation (DLA) software developed by Woźniak (2012) was used, which iteratively adds the primary particle one by one, preserving the fractal parameters at each step.

## 2.2 Description scheme of the simulated BCFAs

Previous modelling studies (Kahnert, 2010; Smith and Grainger, 2014) investigated the optical properties of pure BCFAs i.e., without any coating. From the simulated optical properties, parametrization for pure BCFAs with respect to the number of primary particles at various fractal dimensions and wavelengths were given (Smith and Grainger, 2014). Ouf et al. (2016) conducted Near Edge X-ray Absorption Fine Structure (NEXAFS) analysis on BC produced from a diffusion flame-based mini-CAST burner and found that organics (by-products of the combustion) get attached to the edge of graphite crystallites without changing the inner structure of the core. This laboratory result can be simulated for coated BC in radiative modeling studies by assuming a spherical coating around each individual primary particle of a BC aggregate (Luo et al., 2018). It must be noted that the focus of our study is on BCFAs with coatings consisting of non-absorbing organics. If a brown carbon coating was to be included in the study, information and extra computational time regarding their refractive indices was needed. Unfortunately, due to the time-consuming nature of simulations, the generated database could not include BCFAs with brown carbon coating.

For the sake of simplicity and computational limitations, this representation of coated BC shown in Fig. 2 (bottom panel) was chosen for the entire study. In order to simulate such BC aggregates with individually coated primary particle, the inner radius of the primary particle ($a_i$) is fixed to 15 nm. Whereas the outer radius of the primary particle ($a_o$) consisting of the organics, is varied from 15.1nm to 30nm with the fraction of organics ($f_{organics}$) changing from 1% to 90% respectively. The relationship between the outer radius of the primary particle ($a_o$), the inner radius of the primary particle ($a_i$), and the fraction of organics ($f_{organics}$) is shown below:

$$\frac{4}{3}\pi a_i^3 = (1 - f_{organics})\ \frac{4}{3}\pi a_o^3 \quad . \tag{3}$$

It must be noted that when the fraction of organics ($f_{organics}$) is larger than 80% and the morphology of the aggregate becomes compact, using this coated BC representation results in a practically unrealistic particle (randomly immersed BC primary particles in a spherical coating structure). Therefore, both the composition and morphology of the aggregate play a role while choosing the representation for coated BC. Keeping the above facts in mind, we have limited the use of this coating model only for coated BCFAs with fractal dimension $D_f$ below 2.2. In such cases, where the BC aggregate does not have a completely compact structure, the results are expected to be reliable (Luo et al., 2018). Moreover, Kahnert et al., 2017 compared the coating model (closed-cell model) used in this study to a realistic model, which showed good comparability.

Luo et al., 2018 kept the overall size of aggregates constant to study the sensitivity of optical properties at various number of primary particles ($N_s$) and vice-versa. In our study, the size of the BC aggregates is increased gradually studying the subsequent changes in the optical properties. The optical properties of BC aggregates were calculated for various cases, following a well-designed description scheme summarized in Fig. 1. All the optical properties are calculated at three wavelengths in the visible range i.e., 467nm, 530nm and 660nm. The values are chosen following the availability of refractive index at these specific wavelengths from Kim et al, 2015. For pure BC aggregates, the optical properties were calculated for $1.5 \leq D_f \leq 2.8$ in steps of 0.1. In case of the coated BC aggregates, the optical properties are calculated at the above-mentioned wavelengths for $1.5 \leq D_f \leq 2.2$ in steps of 0.1, and for $1\% \leq f_{organics} \leq 90\%$ in increments of 5%. The approach of assuming a spherical coating around each individual BC primary particle results in an unlikely structure for coated BCFAs with $D_f > 2.2$, hence those cases were omitted in this study. Fig. 2 shows a few of the aggregates from the classification at a fixed $D_f$ and $f_{organics}$. The large dataset obtained from the classification helped in further developing the comprehensive parametrization scheme.

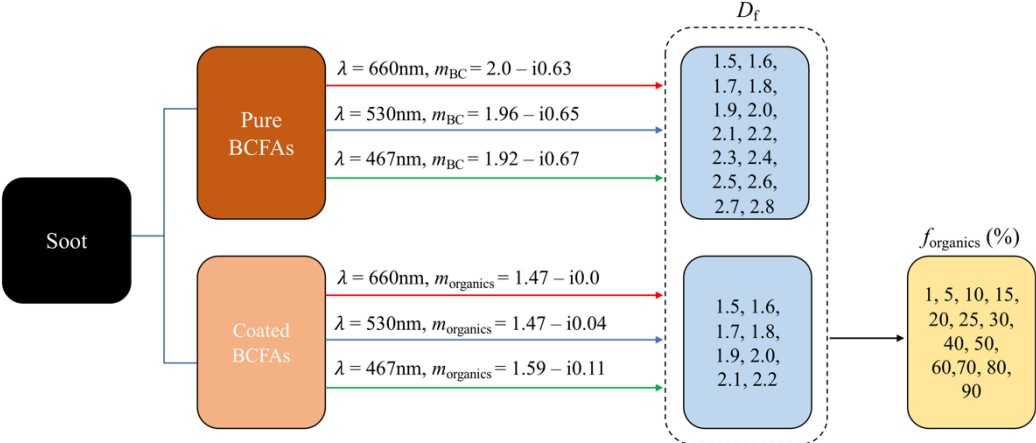

**Figure 1.** The description scheme of black carbon fractal aggregates (BCFAs) adopted in this study.

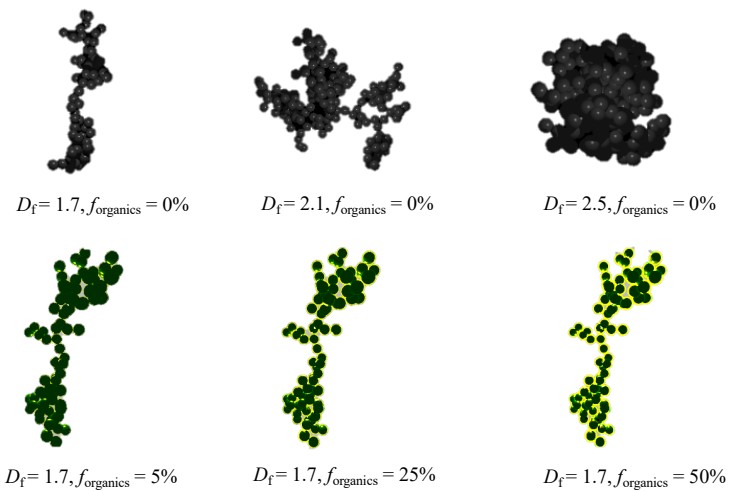

$D_f = 1.7, f_{organics} = 0\%$          $D_f = 2.1, f_{organics} = 0\%$          $D_f = 2.5, f_{organics} = 0\%$

$D_f = 1.7, f_{organics} = 5\%$          $D_f = 1.7, f_{organics} = 25\%$          $D_f = 1.7, f_{organics} = 50\%$

**Figure 2.** Examples of black carbon fractal aggregates (BCFA) with 200 primary particles, and varying $D_f$ and
$f_{organics}$.
In each case of the mentioned classification, the size of the BCFA is changed by incrementing $N_s$ with 5% and
rounded to an integer value, starting from 1 up to 1000. It must be noted that in the results, the size of the BCFA
is expressed in terms of mobility diameter ($D_{mob}$) instead of the number of primary particles ($N_s$) using the simple
conversion developed by Sorensen (2011) given below:
$$D_{mob} = 2a_0(10^{-2x+0.92})N_s^x \tag{4}$$
where, $x$ is the mobility mass scaling exponent given by $x = 0.51Kn^{0.043}$ with $0.46 < x < 0.56$ having an estimated
error of $\pm 0.02$ (Sorensen, 2011). $Kn$ is the Knudsen number, which is the ratio of the molecular free path to the
agglomerate mobility radius.
The conversion formula given in (4) is well founded over the entire range, spanning from the continuum to free
molecular regime. Using pre-calculated values of $x$, the mobility diameter ($D_{mob}$) is derived for the entire dataset.
The relationship between derived mobility diameter ($D_{mob}$), number of primary particles ($N_s$) and volume
equivalent diameter ($D_{equ}$) for a case of pure BCFA with $a_0 = 15$ nm is shown in Fig. 3.

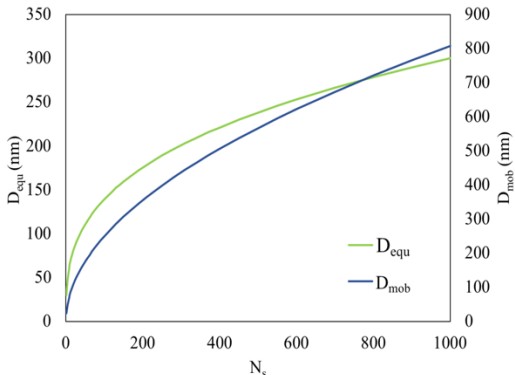

**Figure 3.** Relationship between mobility diameter ($D_{mob}$), number of primary particles ($N_s$) and volume equivalent diameter ($D_{equ}$) for pure BCFAs with $a_o$ = 15nm.

BC has a refractive index fairly wavelength independent in the visible and near-visible spectrum range (Bond and Bergstrom., 2006). There are modelling studies which assume a wavelength independent refractive index of $m = 1.95 + 0.79i$ for BC over the visible spectrum range (Smith and Grainger., 2014; Luo et al., 2018). For organic carbon, the imaginary part of the refractive index ($m_i$) is highly wavelength dependent at the shorter wavelengths in the visible and ultraviolent (UV) wavelengths (Moosmüller et al., 2009; Alexander at al., 2008). Contrary to other studies, Kim et al., 2015 concluded that BC shows a fair amount of wavelength dependency, and provided refractive indices for BC and organics in the visible spectrum. Following his study, the real ($m_r$) and imaginary ($m_i$) part of the refractive indices used for BC and organics at different wavelengths in this study are summarized in table 1.

**Table 1.** Refractive indices ($m_r$ and $m_i$) of BC and organics at various wavelengths in the visible range (Kim et al., 2015) used in this study.

| Parameter | Wavelength (*nm*) | | |
|---|---|---|---|
| | 467 | 530 | 660 |
| $m_{r\_BC}$ | 1.92 | 1.96 | 2.0 |
| $m_{i\_BC}$ | 0.67 | 0.65 | 0.63 |
| $m_{r\_Organics}$ | 1.59 | 1.47 | 1.47 |
| $m_{i\_Organics}$ | 0.11 | 0.04 | 0 |

## 2.3 Optical properties from Multi-Sphere T-matrix Method (MSTM)

Multi-sphere T-matrix Method (MSTM) consists of an algorithm for calculating the time-harmonic electromagnetic properties of a set of arbitrary spheres (Mishchenko et al., 2004; Mackowski and Mishchenko, 2011). The MSTM version 3.0 (Mackowski et al., 2013) calculates the optical properties for fixed and random orientations, the latter being used in this study. MSTM code can calculate the optical properties of coated BCFAs involving nested spheres with the condition that there should be no intersecting surfaces of individual primary particles. Radius, and positions vectors of the inner and outer primary particle of the BCFA are obtained from the tunable DLA software (Woźniak, 2012) which is coupled to the MSTM code.

The optical properties of the aggregates were modelled at three wavelengths, i.e., 467, 530, and 660 nm. At the wavelengths of 660nm and 530nm, the optical properties from MSTM code are obtained for $1 \leq N_s \leq 1000$. Because of the increasing processing time of the MSTM code at lower wavelengths, the calculations are limited to $1 \leq N_s \leq 500$ for a wavelength of 467nm.

For reference purposes, the optical properties were also calculated using the Mie theory, and the absorption cross-section from Rayleigh-Debye-Gans (RDG) theory. For the Mie theory calculations, spheres with volume equivalent radius of aggregates were taken. In case of the coated aggregates, a concentric core-shell configuration was used (He at al., 2015). The RGD theory considers the primary particles in the aggregate as individual Rayleigh scatters, while ignoring the inter-particle scattering (Sorensen, 2001). Therefore, in the RGD theory, the total absorption cross-section of the aggregate ($C_{abs}^{agg}$) is the summation of the absorption cross-sections ($C_{abs}^{pp}$) of

individual primary particles ($N_s$). For a monodisperse distribution, the absorption cross-section from the RDG theory is given as :

$$C_{abs}^{agg} = N_s C_{abs}^{pp}. \tag{5}$$

## 2.4 Optical properties and simplified radiative forcing model

The radiative parameters calculated from the model are briefly presented below. The MSTM code provides the extinction, absorption and scattering efficiency ($Q$), and the asymmetry parameter ($g$) of BCFAs. The extinction, absorption and scattering cross–sections ($C_{ext/abs/sca}$) are further obtained as the product of efficiency ($Q$) and geometric cross-section ($C_{geo}$) by:

$$C_{ext/abs/sca} = (Q_{ext/abs/sca}) * C_{geo}. \tag{6}$$

In spherical objects with radii (R), the geometric cross-section ($C_{geo}$) is related to the radius by:

$$C_{geo} = \pi R^2 . \tag{7}$$

Therefore, for a BCFA, the cross-sections ($C_{ext/abs/sca}$) with volume equivalent radius ($R_v$) are defined as follows:

$$C_{ext/abs/sca} = Q_{ext/abs/sca} \pi R_V^2 . \tag{8}$$

The Volume equivalent radius ($R_v$) is calculated by:

$$R_V = a_o N_s^{\frac{1}{3}}. \tag{9}$$

The single scattering albedo ($SSA$) is the ratio of scattering efficiency ($Q_{sca}$) and extinction efficiency ($Q_{ext}$), where $Q_{ext}$ is the sum of absorption and scattering efficiency as shown below:

$$SSA = \frac{Q_{sca}}{Q_{ext}} = \frac{Q_{sca}}{Q_{sca} + Q_{abs}}. \tag{10}$$

Values of $\omega$ varies from 0 for a purely absorbing particle to 1 for a completely scattering particle.

Mass absorption cross-section (MAC) is calculated from the ratio of absorption cross section ($C_{abs}$) and BC mass ($m_{BC}$) as:

$$MAC = \frac{C_{abs}}{m_{BC}} = \frac{C_{abs}}{\frac{4}{3}\pi R_V^3 \rho_{BC}} , \tag{11}$$

where $\rho_{BC}$ is the density of BC fixed to 1.8 g/cm$^3$ (Bond and Bergstrom, 2006).

The wavelength dependence of light absorption, represented by the Absorption Ångström Exponent (AAE) is calculated using the absorption cross-section ($C_{abs}$) at the three wavelengths ($\lambda$) of 467, 530, and 660 nm. The AAE value is obtained by:

$$C_{abs} (\lambda = 467, 530, 660) = b\lambda^{-AAE} , \tag{12}$$

where $b$ is a constant.

The absorption enhancement factor ($E_\lambda$) is defined by the ratio of absorption cross section of coated BCFA ($C_{abs}^{coated}$) and pure BCFA ($C_{abs}^{pure}$) as shown below:

$$E_\lambda = \frac{C_{abs}^{coated}}{C_{abs}^{pure}}. \tag{13}$$

This implies that the enhancement is given for particles of different total mass but the same BC mass.

To understand the atmospheric implication, the radiative forcing is estimated using a model for absorbing aerosols given by Chylek and Wong, 1995. The black carbon radiative forcing at the top of the atmosphere is calculated as:

$$\Delta F_{TOA} = -\frac{S_o}{4}(1 - N_{cloud})T^2 2\tau[(1 - a)^2\beta\omega - 2a(1 - \omega)] \tag{14}$$

where, $S_o$ is the solar constant, $N_{cloud}$ is the cloud fraction, $T$ is the transmittance of the sky above the layer of
aerosols, $\tau$ is the aerosol optical depth, $\beta$ is the upward scattering function, $a$ is the surface albedo, and $\omega$ is the
single scattering albedo. From Sagan and Pollack, 1967, the upward scattering function $\beta$ is calculated from the
asymmetry parameter $g$ by:

$$\beta = \frac{1}{2}(1 - g) \tag{15}$$

The model given by Chylek and Wong (1995) for the calculation of TOA forcing is a simplified version of the
multiple reflection model (Haywood and Shine, 1995; Sheridan and Ogren, 1999) with some implicit
approximations. It is important to note that this is an analytical model which can be useful to understand the
sensitivities of radiative forcing to various parameters (Chylek and Wong, 1995; Lesins et al., 2002). The
simplified version was used in this study to highlight the sensitivity of the TOA forcing towards the morphology
and composition of BC. However, the model cannot be used to replace the accurate direct radiative forcing
calculations.

**3    Results and discussion**

**3.1   Variability in optical properties due to randomized particle generation**

In the tunable DLA program, the user specified values of number of spheres ($N_s$), radius of the primary particle
($a_o$), and fractal dimension ($D_f$) are used to generate the fractal aggregate. This gives rise to a possibility of more
than one representation of a fractal aggregate satisfying the same fractal dimension ($D_f$) i.e., randomized particle
generation. The difference between the various representations being only the different positions of the primary
particles constituting the aggregate. This further results in an uncertainty in the radiative results. Depending on
the complexity, some studies averaged the radiative results over 5-10 representations (Wu et al., 2016; Luo et al.,
2018), whereas others consider only a single representation (Smith and Grainger, 2014).
Considering the large dataset in this study, the option of taking an average of the multiple representations would
be time-consuming. Therefore, the general uncertainty in optical properties for 30 representations of the pure
BCFAs is discussed. This is done for various cases of size ($D_{mob}$) and morphology ($D_f$). Fig. 4 shows the variability
in the extinction cross-section $C_{ext}$ (first row), absorption cross-section $C_{abs}$ (second row), scattering cross-section
$C_{sca}$ (third row), and asymmetry parameter $g$ (fourth row) as a function of $D_f$. The results were calculated at a
wavelength of 660 nm for pure BCFAs of $D_{mob}$ values 150nm, 250nm, 500nm, and 1000nm increasing from left
to right in the Fig. 4.
The uncertainty in the optical properties was studied for 30 representations of BCFAs with the same value of
the fractal dimension. The amount of variability in the optical property at each fractal dimension (x-axis) must be
seen from the whiskers of the boxplot in Fig. 4. The sensitivity of the optical properties with respect to various
fractal dimensions can be figured out from the amount of overlapping of the y-axis values between adjacent
boxplots.
For extinction and scattering cross-sections (first and third row), the uncertainty is more pronounced at $D_f <$
1.7. This is because of the overlapping of extinction and scattering cross-sections values at $D_f < 1.7$. The
absorption cross-section ($C_{abs}$) shows the highest uncertainty towards various representations of a BCFA which
can be seen from higher heights of boxplots in panel (e), (f), and (g) of the Fig. 4. Additionally, at 150 nm and
250 nm, $C_{abs}$ is seen to be less sensitive towards $D_f$ ranging between 1.5 - 2. Whereas, for boxplots in panel (g)
representing a 500nm BCFA, the $C_{abs}$ values overlap for $D_f > 1.8$. It may be noted that the $C_{abs}$ increases with $D_f$
for smaller BCFA (panel (e) and (f)), whereas the opposite is true for larger BCFA (panel (g) and (h)) as also
reported by Luo et al, 2018. This is further explained in detail in the section 3.3. The asymmetry parameter ($g$)
shows a similar uncertainty trend to that of the extinction and scattering cross-sections i.e., lower variability but
some overlapping at certain $D_f$ seen in fourth row. In general, it is observed that the uncertainty of optical
properties at larger sizes ($D_{mob}$ =1000nm; last column) is comparatively low. The standard deviation in the optical
properties is averaged over size, and summarized for various cases of $D_f$ in Table 2.

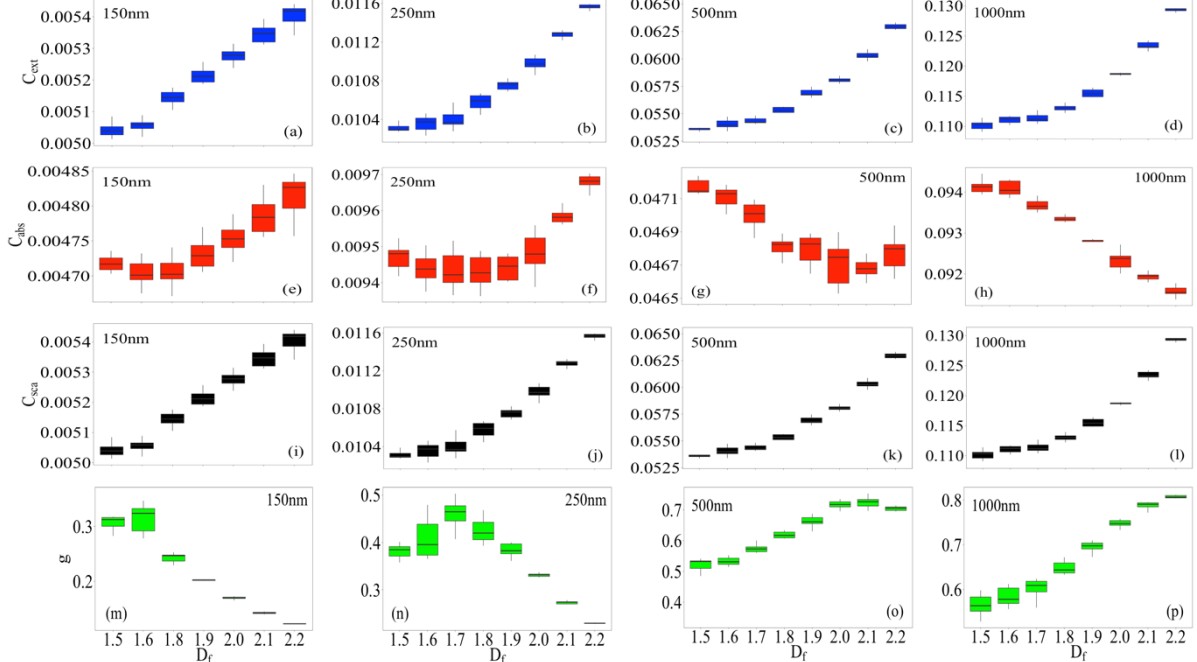

**Figure 4.** The variability in the optical properties at $\lambda = 660$nm for 30 representations of pure BCFAs with $D_{mob}$
increasing (left to right). The panels show extinction cross-section $C_{ext}$ (first row), absorption cross-section $C_{abs}$
(second row), scattering cross-section $C_{sca}$ (third row), and asymmetry parameter $g$ (fourth row). The boxplots
show the interquartile range between 75 - 25 percentile, with the center bar in the box indicating the median. The
whiskers on the top and bottom of the boxplot mark the largest and smallest value within 1.5 times interquartile
range.
**Table 2.** Summary of the variability (%) in the optical properties of pure BCFAs. The variability of extinction
cross-section $C_{ext}$, absorption cross-section $C_{abs}$, scattering cross-section $C_{sca}$, asymmetry parameter $g$, and single
scattering albedo SSA are shown for fractal dimension ($D_f$) between 1.5 to 2.2. For each case, the resultant
variability is an average over the sizes of 100, 250, 500, and 1000nm.

| Optical property | Fractal dimension ($D_f$) | | | | | | | |
|---|---|---|---|---|---|---|---|---|
| | 1.5 | 1.6 | 1.7 | 1.8 | 1.9 | 2 | 2.1 | 2.2 |
| $C_{ext}$ | 0.54 | 0.75 | 0.65 | 0.56 | 0.54 | 0.46 | 0.73 | 0.73 |
| $C_{abs}$ | 0.24 | 0.26 | 0.34 | 0.24 | 0.20 | 0.39 | 0.36 | 0.36 |
| $C_{sca}$ | 4.68 | 5.90 | 4.68 | 3.25 | 2.68 | 1.52 | 2.97 | 2.97 |
| $g$ | 5.81 | 5.24 | 4.32 | 2.90 | 1.76 | 1.45 | 3.36 | 1.56 |
| SSA | 4.20 | 5.29 | 4.09 | 2.71 | 2.17 | 1.17 | 2.29 | 2.29 |

**3.2   Optical properties of BCFAs at different radius of the primary particle**
The absorption cross-section ($C_{abs}$) and BC mass absorption cross-section ($MAC_{BC}$) have been reported to be
insensitive to radius of the primary particle ($a_o$) for a fixed particle volume (Kahnert, 2016b). Fig. 5 shows the
optical properties of pure BCFAs with the radius of primary particle ($a_o$) varying between 15nm and 30nm as a
function of $D_{mob}$. The results were calculated for a wavelength of 660nm for pure BCFAs with $D_f = 1.7$. The
absorption cross-section $C_{abs}$ showed in panel (b) increases by a factor of almost ten from $a_o$ equal to 15nm to
30nm. Since our results here are represented against $D_{mob}$ instead of volume equivalent radius ($R_{equ}$), they are not
expected to follow the findings of Kahnert, 2016b. The results with respect to the $R_{equ}$ are provided in the Fig. S1
of the supplementary material, which follow the findings of Kahnert, 2016b. The asymmetry parameter shows the
least dependency on $a_o$ as can be seen in panel (d). The single scattering albedo (SSA) and the BC mass absorption
cross-section ($MAC_{BC}$) shown in panel (e) and (d) of the Fig. 5 show a larger increase at $a_o > 20$nm for the same
$D_{mob}$. Acknowledging the effect of changing $a_o$ over the optical properties, for the sake of simplicity, in this study
the inner radius of the primary particle ($a_i$) was fixed to 15nm, and the outer radii of the primary particle ($a_o$) was
increased with $f_{organics}$.

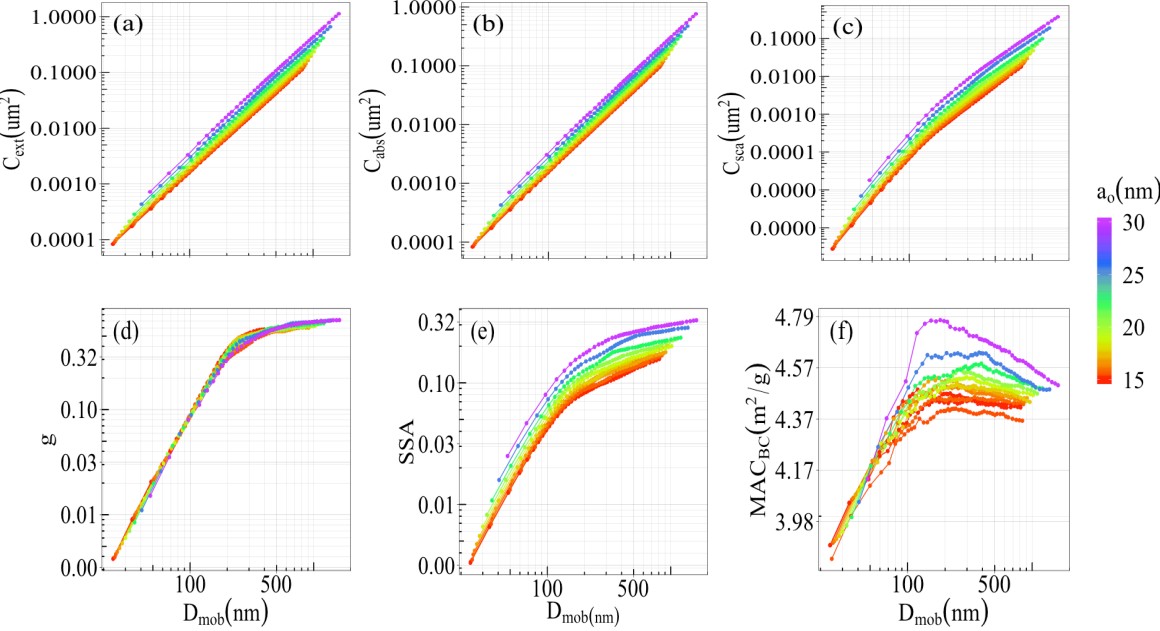

**Figure 5.** Optical properties of pure BCFAs at various radii of primary particle ($a_o$) with respect to mobility
diameter ($D_{mob}$): (a) extinction cross-section $C_{ext}$, (b) absorption cross-section $C_{abs}$, (c) scattering cross-section
$C_{sca}$, (d) asymmetry parameter $g$, (e) single scattering albedo $SSA$, and (f) black carbon mass absorption cross-
section $MAC_{BC}$ at $\lambda = 660$nm.

### 3.3 Dependency of BCFA optical properties on the morphology

Different optical properties as a function of changing $D_{mob}$, and $D_f$ are shown in Fig. 6. The results were calculated
for pure BCFAs ($f_{organics} = 0$) at a wavelength of 660nm. The cross-sections (panel (a), (b), and (c)) show an
increase with $D_{mob}$. The cross-sections vary from 0.0001$\mu m^2$ to 0.1$\mu m^2$ for BCFA $D_{mob}$ ranging from 24nm to
810nm. The extinction and scattering cross-sections are larger for higher $D_f$, suggesting an increasing coherent
scattering for compact morphologies also reported by Smith and Grainger (2014). The dependency of the optical
cross-section over the fractal dimension ($D_f$) was pointed out by Berry and Percival (1986) where the change in
the cross-sections depends on whether the fractal dimension ($D_f$) is less than two or greater than two. The results
from Mie calculations for a spherical particle ($D_f = 3$) follows the trend of the MSTM results as seen in the Fig.
6.
For smaller BCFAs, the absorption cross-section shows negligible dependence on $D_f$. With increasing size, the
absorption cross-section decreases with $D_f$. This decrease can be interpreted as a shielding effect due to the
primary particles on the surface of the aggregate. Further, with $D_f > 2.5$, the absorption cross-section increases
with $D_f$ showing the highest value for a spherical particle ($D_f = 3$). This may be caused by Mie resonances in larger
BCFAs. Earlier studies have also reported higher values for the sphere equivalent ($D_f = 3$) calculations of BCFA
(Liu et al., 2018; Li et al., 2016).
The single scattering albedo (SSA = $C_{sca}/C_{ext}$) shown in panel (e) of Fig. 6 has values up to 0.42. The SSA also
increases with $D_{mob}$ and $D_f$, the latter is explained by the decreasing scattering in loosely packed BCFAs. This is
due to compact aggregates following a Rayleigh-like polarization curve (Gustafson and Kolokolova, 1999). The
asymmetry parameter ($g$) (panel d) shows a range of values between 0 and 1 for values of $D_{mob}$ values between
24nm and 810nm. The asymmetry parameter is higher for chain-like BCFAs with lower $D_f$, indicating larger
forward scattering in asymmetrical structures also reported by Luo et al. 2018. When the BCFAs grow larger. the
rate of increase in $g$ with size gradually decreases for lower $D_f$ because of the scattering tending to the Rayleigh
scattering regime.
Black carbon mass absorption cross-section ($MAC_{BC}$) values shown in panel (f) fall within the range of findings
reported in the literature (Bond and Bergstrom, 2006). The $MAC_{BC}$ increase with $D_{mob}$ showing a peak at $D_{mob} \sim$
250nm. The dependency of $MAC_{BC}$ on $D_f$ is similar to that of the absorption cross-section i.e., Mie resonances
contribute to the increase at higher $D_f$, explaining the large discrepancy between MSTM and Mie results for
$MAC_{BC}$. The above results with respect to the $R_{equ}$ are provided in the Fig. S2.

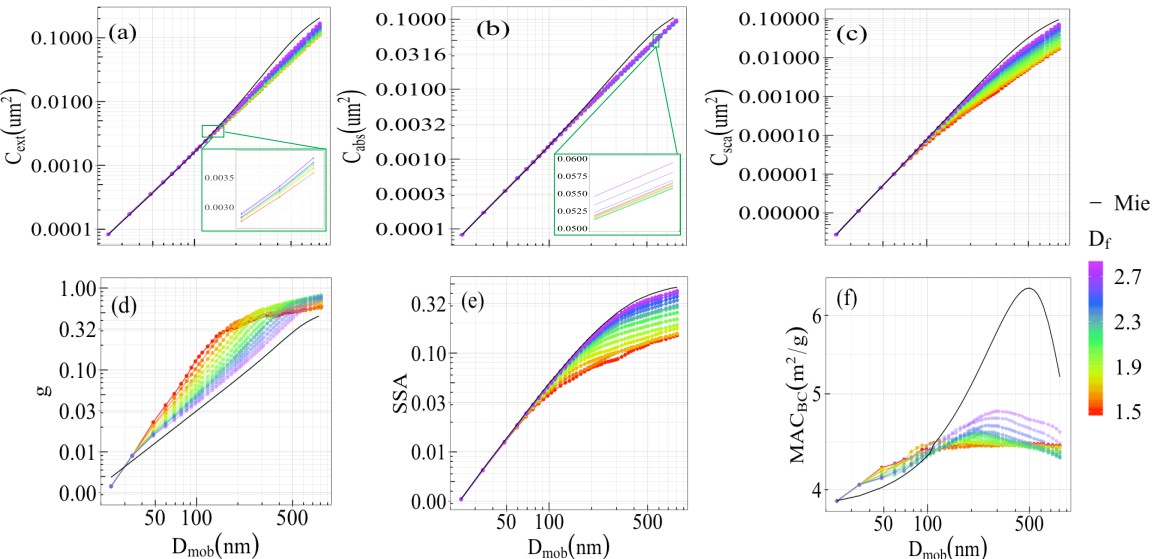

**Figure 6.** Optical properties of pure BCFAs as a function of $D_{mob}$ at various fractal dimension ($D_f$): (a) extinction
cross-section $C_{ext}$, (b) absorption cross-section $C_{abs}$, (c) scattering cross-section $C_{sca}$, (d) asymmetry parameter $g$,
(e) single scattering albedo $SSA$, and (f) black carbon mass absorption cross-section $MAC_{BC}$ at $\lambda$ = 660nm.
Radiative results from the Mie calculations are shown by the black line (panel a-f). The $C_{abs}$ from the Rayleigh-
Debye-Gans (RDG) theory is represented by a dash line (panel b).
**3.4   Dependency of BCFA optical properties on $f_{organics}$**
Figure 7 shows how the optical properties of BCFAs are influenced by the increasing content of organics. The
calculations were done for a BCFA of chain-like morphology with $D_f$ =1.7 at a wavelength of 660nm. The results
are shown as function of $D_{mob}$ at various fractions of organics ($f_{organics}$). The extinction and absorbing cross-sections
(Fig. 7a and 7b) decrease steadily with increasing $f_{organics}$ for constant mobility diamters because of the increasing
less-absorbing volume fraction in the aggregate. The dependence on the asymmetry parameter $g$ (Fig. 7d) on
$f_{organics}$ is very small, meaning that $g$ is more sensitive to morphology rather than composition. The single scattering
albedo (SSA) increases with $f_{organics}$, and this is again because of the increasing fraction of less absorbing material.
From the results of black carbon mass absorption cross-section ($MAC_{BC}$) values shown in Fig. 7f, a dominating
dependence of BCFA on composition is seen, in comparison to size and morphology. Similar results for a compact
BCFA of $D_f$ =2.2 at a wavelength of 660nm can be found in the Fig. S4 of the supplementary material.
Figure 8 is similar to Fig.6 but shows the dependency of optical properties on the fractal dimension ($D_f$) for
organic coated BCFAs with $f_{organics}$ of 50% at the wavelength of 660nm. The cross-sections and asymmetry
parameter show similar behaviour such as that of the pure BCFAs. The SSA has an upper limit of 0.35 at $D_f$=2.2.
Black carbon mass absorption cross-section ($MAC_{BC}$) is rather independent of $D_f$ but values increase with coating
by a factor of 1.2 for coated BCFAs with $f_{organics}$ of 50% as shown in Fig. 7.
The gradually decreasing impact of the fractal morphology on the optical properties of coated BC particles was
shown by Liu et al., 2017. In this study, it is seen in the case of a non-coated BC particle (Fig .6c), the $C_{sca}$ is more
sensitive to the $D_f$, whereases, when the BC particles are coated (Fig. 7c, Fig 8c), the $C_{sca}$ is less sensitive towards
$D_f$ and $f_{organics}$. It is observed that the $C_{sca}$ and $SSA$ (Fig. 8c, Fig. 8e) become more sensitive to $D_f$ when the BCFA
grows in size, therefore, the impact of the fractal morphology over the optical properties is also a function of
particle size.  Moreover, it must be noted that even though there is a decreasing impact of the fractal morphology
on optical properties, parameters like $C_{abs}$, $MAC_{BC}$, and $g$ showed significant variability towards changes of $f_{organics}$
(Fig 7a, 7b, 7e, and 7f).
Global models use Mie theory for calculations of BC optical properties (Bond et al., 2013). The Mie theory
considers BC as homogeneously mixed spheres, or as a core-shell configuration. The results of $SSA$, $g$, and $MAC_{BC}$
in both Fig.6 and Fig.8 clearly demonstrate a significant influence of morphology. This is clearly seen from the
difference between the coloured lines representing various morphologies of BC as aggregates, and the black solid
line representing the result when BC is assumed as a core-shell. Therefore, the factor of changing morphology is
not considered adequately when using the Mie theory for BC optical properties in global models.

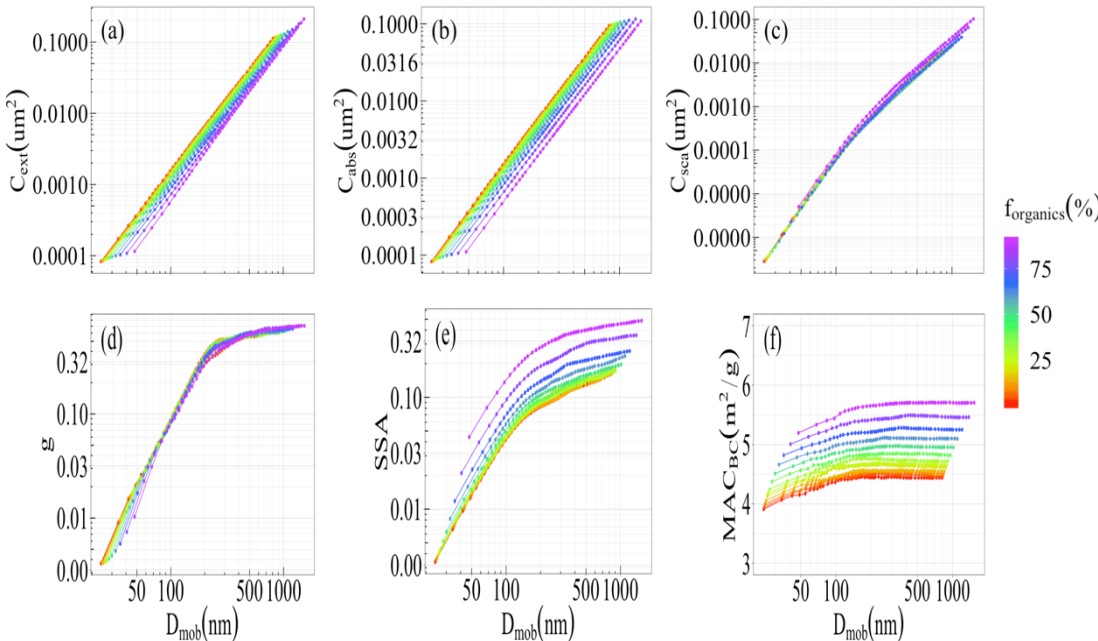

**Figure 7.** Optical properties of BCFAs ($D_f = 1.7$) as a function of $D_{mob}$ at various fraction of organics ($f_{organics}$):
(a) extinction cross-section $C_{ext}$, (b) absorption cross-section $C_{abs}$, (c) scattering cross-section $C_{sca}$, (d) asymmetry
parameter $g$, (e) single scattering albedo $SSA$, and (f) black carbon mass absorption cross-section $MAC_{BC}$ at $\lambda =$
660n

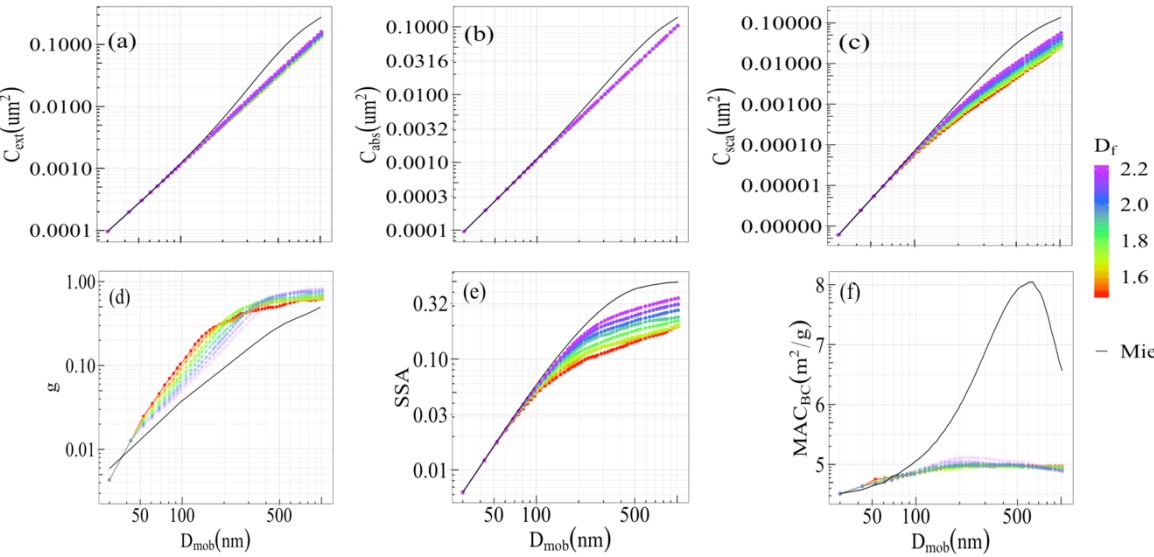

**Figure 8.** Optical properties of coated BCFAs ($f_{organics} = 50\%$) as a function of $D_{mob}$ at various fractal dimension
($D_f$): (a) extinction cross-section $C_{ext}$, (b) absorption cross-section $C_{abs}$, (c) scattering cross-section $C_{sca}$, (d)
asymmetry parameter $g$, (e) single scattering albedo $SSA$, and (f) black carbon mass absorption cross-section
$MAC_{BC}$ at $\lambda = 660$nm.
### 3.5 Dependency of BCFA optical properties on wavelength
In the sections before, the dependency of BCFA optical properties on size, morphology, and composition were
discussed. In this section, besides showing the spectral dependency of BCFA optical properties, it is also
demonstrated how this dependency changes with morphology, and composition in the visible wavelength range.

Figure 9 shows the changes in the pure BCFAs optical properties with wavelength ($\lambda$) at various morphologies represented by $D_f$. Pure BCFAs with fixed $D_{mob}$ equal to 330nm were taken for this case to demonstrate the effect of morphology. All the optical properties show a decrease with $\lambda$ in the visible range. Furthermore, it was studied whether the rate of decrease might vary for various morphologies. Fig. 9 show that the spectral dependency is insensitive to morphology for the absorption cross-section $C_{abs}$ (panel (b)) and black carbon mass absorption cross-section $MAC_{BC}$ (panel (f)). The spectral dependence of scattering cross-section $C_{sca}$ (panel (c)) is seen to be somewhat sensitive towards changes in morphology. The highest sensitivity of spectral dependence to morphology was seen for the asymmetry parameter ($g$), dominant at higher $D_f$ i.e., for compact aggregates.

Figure 10 is provided to illustrate how the spectral dependency of BCFAs changes with composition i.e., fraction of organics ($f_{organics}$). For this case, BCFAs are considered with $N_s$ and $D_F$ equal to 200 and 1.7 respectively. It must be noted that the size of the BCFAs is also increasing with $f_{organics}$. Contrary to the results from Fig. 9, all the cross-sections (panel (a), (b), and (c)) and black carbon mass absorption cross-section $MAC_{BC}$ (panel (f)) show a significant increase in the spectral dependency with $f_{organics}$. The spectral dependency of single scattering albedo SSA (panel (d)) shows a comparatively lower sensitivity towards $f_{organics}$, whereas it's nearly negligible for the asymmetry parameter ($g$) seen in panel (e). Additionally, the change in spectral dependency over the size is also shown in the Fig. S5 of the supplementary material.

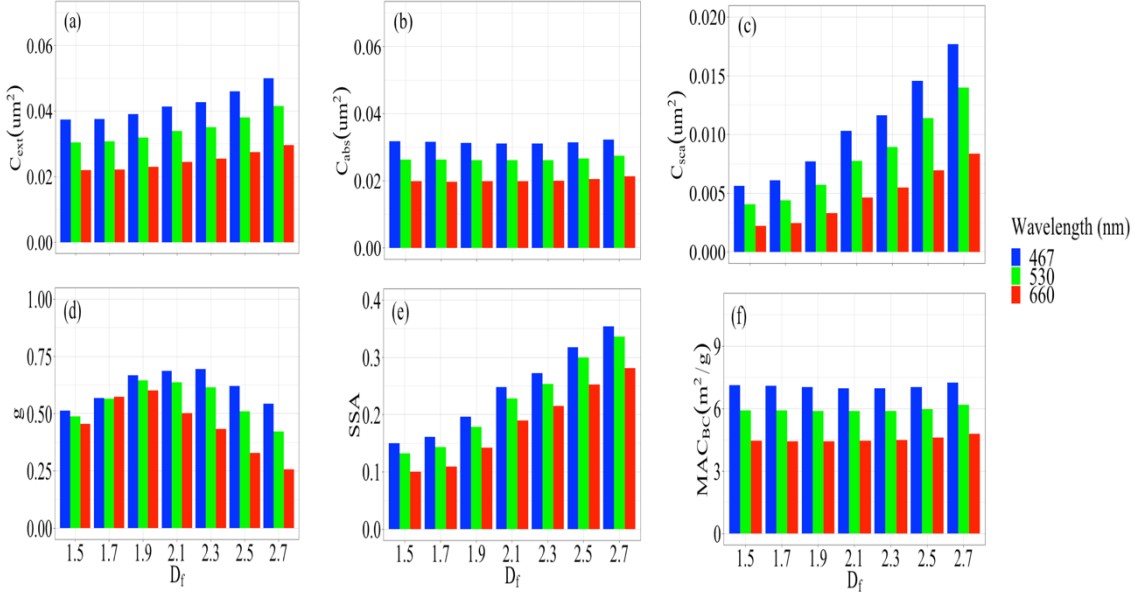

**Figure 9.** Spectral dependency of the pure BCFAs optical properties ($D_{mob}$ = 330nm) on fractal dimension ($D_f$): (a) extinction cross-section $C_{ext}$, (b) absorption cross-section $C_{abs}$, (c) scattering cross-section $C_{sca}$, (d) asymmetry parameter $g$, (e) single scattering albedo *SSA,* and (f) black carbon mass absorption cross-section $MAC_{BC}$. For the variability (%) in different cases of $D_f$ refer to Table 2.

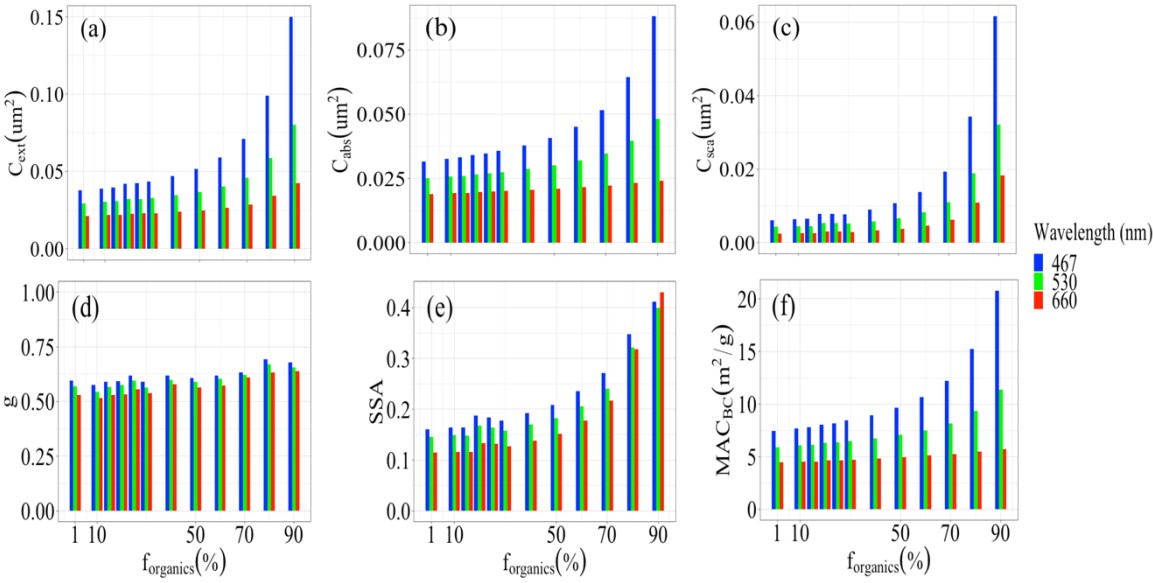

**Figure 10.** Spectral dependency of coated BCFAs optical properties ($N_s$ = 200, $D_f$ = 1.7) on fraction of organics
($f_{organics}$): (a) extinction cross-section $C_{ext}$, (b) absorption cross-section $C_{abs}$, (c) scattering cross-section $C_{sca}$, (d)
asymmetry parameter $g$, (e) single scattering albedo *SSA,* and (f) black carbon mass absorption cross-section
$MAC_{BC}$. For the variability (%) refer to the case $D_f$ = 1.7 in Table 2.

**3.6 Ångström absorption exponent (AAE) and enhancement factors ($E_\lambda$)**
Figure 11 shows the Ångström absorption exponent (AAE) of a chain-like BCFA ($D_{f\,=}$ 1.7) as a function of
mobility diameter ($D_{mob}$), and increasing fraction of organics ($f_{organic}$). The AAE is derived from the slope of $C_{abs}$
vs $\lambda$ at 467, 530, and 660 nm as shown in Eq. (12). As expected, the AAE shows a straightforward dependency
on the fraction of organics ($f_{organic}$). In this case, the values of AAE vary from 1.4 up to 3.6 with increase in $f_{organic}$
from 1% until 90%. The variability in the modelled values of AAE may be attributed to the selection of the
refractive indices and wavelengths (Liu et al., 2018). Similar result for the Ångström absorption exponent (AAE)
of a more compact BCFA ($D_{f\,=}$ 2.2) is provided in the Fig. S6. Additionally, the impact of morphology or fractal
dimension ($D_f$) on the AAE for pure BCFAs is shown in Fig. 12 with values ranging from 1.06 to 1.47. It can be
observed that for smaller BCFAs, the AAE increases as the BCFA becomes more compact, whereas for larger
BCFA an opposite effect is seen. Fig. 11 and 12 could be interpreted as the ageing process of BC in the atmosphere
focusing on changing composition and shape respectively.
Figure 13 shows the trend in absorption enhancement factors ($E_\lambda$) as a function of mobility diameter ($D_{mob}$) and
increasing fraction of organics ($f_{organic}$) for a BCFA with $D_{f\,=}$ 1.7. The top row shows the absorption enhancement
factors calculated from the results of the MSTM code ($E_{MSTM}^\lambda$) whereas, the ones derived from the Mie
calculations ($E_{Mie}^\lambda$) are displayed in the bottom row. In general, the Mie derived absorption enhancement factors
are larger by a factor of 1.1 to 1.5. The enhancement results from both MSTM and Mie calculations are shown
for three wavelengths i.e., 660, 530, and 467nm (right to left). There is an expected increase in the absorption
enhancement factors as the wavelength decreases. The values of the modelled absorption enhancement factors
follow the results from various ambient studies which measured enhancement factors ranging from 1.0 to 2.25 at
wavelengths between 532nm to 678nm (Cappa et al., 2012; Cui et al., 2016; Wu et al., 2018).
Liu et al., 2017 emphasized the role of the mass ratio of non-BC to BC on the performance of various methods
used for simulating the scattering cross-section and enhancement factors of BC particles. In this study, it is shown
that the Ångström absorption exponent (AAE) calculated from just the MSTM method can show variability of up
to a factor of two with an increasing non-BC mass fraction larger than 90%. Similarly, it can be seen that the
difference in the enhancement factors calculated from the core-shell theory and fractal assuming MSTM method
can be up to by values between of 1.1 and 1.5.

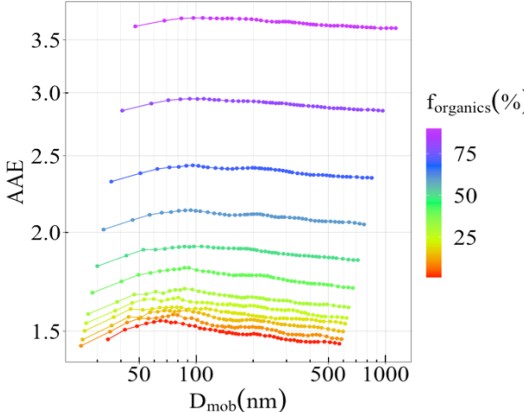

**Figure 11.** Ångström absorption exponent (AAE) of coated BCFAs ($D_f$ = 1.7) with changing fraction of organics
($f_{organics}$) and mobility diameter ($D_{mob}$).

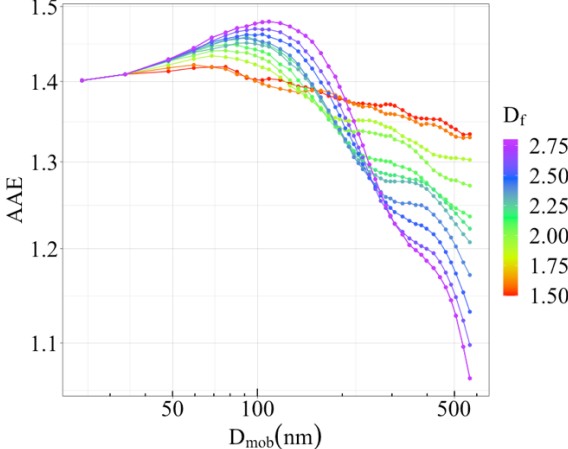

**Figure 12.** Ångström absorption exponent (AAE) of pure BCFAs ($f_{coating}$ = 0%) with changing fractal dimension
($D_f$) and mobility diameter ($D_{mob}$).

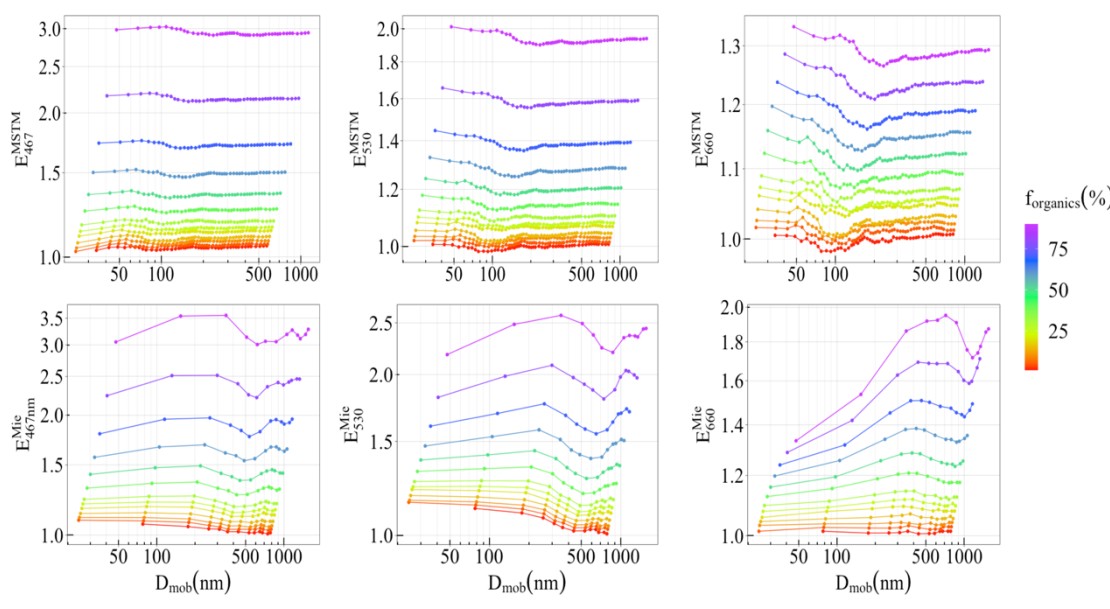

**Figure 13.** Absorption enhancement factor ($E_\lambda$) in BCFAs with $D_f$ = 1.7, changing fraction of organics ($f_{organics}$)
and mobility diameter ($D_{mob}$). The top row shows the $E_\lambda$ derived from the MSTM method whereas the ones derived

from Mie code are shown in the bottom row. The enhancement factors are shown for wavelengths equal to 660, 530, and 467nm (right to left).

### 3.7 Implications of morphology and composition over black carbon radiative forcing

In this section, the dependence of the black carbon radiative forcing on modifying composition and morphology of BCFAs is discussed. The relative changes in the top of the atmosphere radiative forcing ($\Delta F_{TOA}$) are quantified as a function of fractal dimension ($D_f$) and fraction of organics ($f_{organics}$). It is a sensitivity analysis, applicable mostly to scenarios with urban pollution having a high mass fraction of combustion aerosols. The black carbon radiative forcing at the top of the atmosphere ($\Delta F_{TOA}$) is estimated using equation (14) with fixed values of $S_o = 1368$ Wm$^{-2}$, $N_{cloud}$=0.6, $T = 0.79$, $\tau = 0.03$, and $a = 0.1$ (Chylek and Wong, 1995; Lesins et al., 2002). To focus primarily on radiative effects of BC, the optical depth $\tau$ is taken as 0.03 for smoke aerosol (Penner et al., 1992). The values of $\beta$ and $\omega$ change with fractal dimension ($D_f$) and fraction of organics ($f_{organics}$), and are obtained from the MSTM bulk optical properties. The bulk optical properties are calculated at a wavelength of 530 nm, over a lognormal polydisperse size distribution with the geometric mean radius ($r_o$) and standard deviation ($\sigma$) fixed to 0.12μm and 1.5, respectively. The details about the bulk optical properties can be found in the supplementary material of this work.

Table. 3 shows how the values of black carbon radiative forcing change for various morphologies represented by fractal dimension ($D_f$) for pure black carbon. This can be further understood by the relative change C defined by:

$$C = \frac{\Delta F_{TOA} - \Delta F_{TOA}^{Ref}}{\Delta F_{TOA}^{Ref}} \times 100 \tag{16}$$

where $\Delta F_{TOA}^{Ref}$ is the top of the atmosphere radiative forcing for a reference case where the fractal dimension ($D_f$) is 1.7 i.e., a freshly emitted black carbon particle.

**Table 3.** Black carbon radiative forcing $\Delta F_{TOA}$ (Wm$^{-2}$) calculated for various fractal dimension ($D_f$) and relative change ($C$) with respect to a reference case with $D_f$ =1.7.

| $D_f$ | $\Delta F_{TOA}$ | $C$ (%) |
|---|---|---|
| 1.5 | 0.704 | -1.1 |
| 1.6 | 0.721 | -2.3 |
| 1.8 | 0.697 | -3.4 |
| 1.9 | 0.681 | -5.6 |
| 2 | 0.649 | -9.9 |
| 2.1 | 0.608 | -15.7 |
| 2.2 | 0.581 | -19.4 |
| 2.3 | 0.570 | -21.0 |
| 2.4 | 0.507 | -29.7 |
| 2.5 | 0.446 | -38.2 |
| 2.6 | 0.383 | -46.9 |
| 2.7 | 0.324 | -55.1 |
| 2.8 | 0.279 | -61.2 |

Similarly, the values of the black carbon radiative forcing for various compositions represented by fraction of organics ($f_{organics}$) in a case where the fractal dimension ($D_f$) is fixed to 2.2 is shown in Table. 4. The values of relative change ($C$) are calculated using equation (16) with respect to $\Delta F_{TOA}^{Ref}$ as a reference case of zero fraction of organics ($f_{organics}$) i.e., pure black carbon particle.

Global models use the simplified core-shell representation for BC (Bond et al., 2013) which is morphologically close to a coated BCFA of $D_f$ 2.8. In the case of coated BCFA, there is a relative change ($C$) of 20% when $D_f$ increases from 1.5 to 2.2. Following the results in Table. 4 the relative change ($C$) in $\Delta F_{TOA}$ of coated BCFA is also expected to increase as the $D_f$ approaches 2.8. Therefore, the influence of morphology over the $\Delta F_{TOA}$ is considered adequately when using the simplified core-shell representation of BC.

It can be seen from from Table 4 that the top of the atmosphere forcing $\Delta F_{TOA}$ decreases by up to 55% as the organic content of the particles increases to 90%. This result is in agreement with the findings of Zeng et al., 2019 where the increasing hygroscopicity of the BC particle results in negative top of the atmosphere forcing. However, it must be noted that in the study of Zeng et al., 2019, the focus was over aged BC particles with 90-99% coating fraction and the Santa Barbara DISORT Atmospheric Radiative Transfer (SBDART) model was used for estimating the radiative forcing.

Even though the simplified radiative model for absorbing aerosols used, the results of relative change (C) in Table 3 and Table 4 can provide insights about the implications of BC ageing on their radiative forcing estimates. It is demonstrated that the radiative forcing results are highly sensitive towards modifications in morphology and composition when using the aggregate representation. It must be noted that these results are of high relevance in the BC hotspots regions of Asia, for example, Manilla in Philippines, where the BC emission shared up to 70% of calculated PM$_1$ (particulate matter with diameter < 1µm) mass emission factors (Madueno et al., 2019).

**Table 4.** Black carbon radiative forcing $\Delta F_{TOA}$ (Wm$^{-2}$) calculated for various fractions of organics ($f_{organics}$) and relative change (C) with respect to a reference case with $f_{organics}$ = 0%.

| $f_{organic}$ | $\Delta F_{TOA}$ | $C$ (%) |
|---|---|---|
| 1 | 0.581 | -1.6 |
| 5 | 0.572 | -1.5 |
| 10 | 0.572 | -2.4 |
| 15 | 0.567 | -1.6 |
| 20 | 0.572 | -2.4 |
| 25 | 0.567 | -1.5 |
| 30 | 0.572 | -2.3 |
| 40 | 0.568 | -5.1 |
| 50 | 0.552 | -10.0 |
| 60 | 0.523 | -12.8 |
| 70 | 0.507 | -19.0 |
| 80 | 0.471 | -32.8 |
| 90 | 0.391 | -54.6 |

### 3.8 Parametrization scheme for coated BCFAs

In this section, the optimal fits for the results of the optical properties obtained from the MSTM code are discussed. For the extinction and absorption cross-sections, a first order polynomial on logrithmic scales was found to be the best fit.

$$lnC_{ext} = c_0 + c_1 lnD_{mob} \tag{17}$$

$$lnC_{abs} = g_0 + g_1 lnD_{mob} \tag{18}$$

For the fittings of scattering cross-section ($C_{sca}$) and SSA, an equation of the following form was found to fit best. The asymmetry parameter ($g$) is well captured by a cubic polynomial in a logarithm space of $D_{mob}$.

$$lnC_{sca} = H_0 + H_1 lnD_{mob} + H_2 ln(lnD_{mob}) \tag{19}$$

$$lnSSA = k_0 + k_1 lnD_{mob} + k_2 ln(lnD_{mob}) \tag{20}$$

$$lng = \sum_{n=0}^{3} s_n lnD_{mob}^n \tag{21}$$

Since the nature of the curve for mass absorption cross-section ($MAC_{BC}$) changes for various $D_f$, it was not possible to find an optimal function representative for the entire dataset. For all the fits, a limitation was found that the smaller particles are not well represented by the above-mentioned functions. Therefore, in order to find an overall

good fit, the data is taken for points with $D_{mob}$ larger than 50nm. Previous studies have also attempted to fit the
optical properties of pure BCFAs with respect to the number of primary particles ($N_s$) (Smith and Grainger, 2014;
Kahnert, 2012b).
In this study, the parametrization scheme is developed for five BC optical properties, the extinction cross-
section $C_{ext}$, absorption cross-section $C_{abs}$, scattering cross-section $C_{sca}$, single scattering albedo $SSA$, and
asymmetry parameter $g$ with respect to BC size. In total, the fit coefficients for the five BC optical properties are
provided for 192 cases comprising of various combinations of wavelengths ($\lambda$), fractal dimensions ($D_f$) and
fraction of organics ($f_{organics}$) shown in Fig. 1. For each case, linear regression models were applied individually to
the MSTM modelled optical properties for BC sizes ranging from 10 to 1000nm. The fit coefficients for the five
optical properties in each case are provided in a tabular form as a supplement to this work.
The resultant parametrization scheme provides the user an option to estimate the five optical properties at
desired BC size for any of the 192 combinations of $\lambda$, $D_f$, and $f_{organics}$. It must be noted that the MSTM modelled
optical properties were calculated for fixed values of refractive index because of limited computational resources.
Therefore, the parametrization scheme provided in this study is not able to account for variable refractive indices.

### 3.8.1  Error analysis of the parametrization scheme

In this scheme, the parametrization for optical properties of BCFAs are provided for each point of the classification
given in Fig. 1. In the case of pure BCFAs, the parametrization is provided for all combinations of $\lambda$ (nm)= {467,
530, 660}, and $D_f$ = {1.5, 1.6, 1.7, 1.8, 1.9, 2.2, 2.1, 2.3, 2.4, 2.5, 2.6, 2.7, 2.8}. Whereas, in the coated BCFAs,
the parametrization scheme is available for combinations of $\lambda$ (nm) = {467, 530, 660}; $D_f$ = {1.5, 1.6, 1.7, 1.8,
1.9, 2.2} and $f_{organics}$ (%) = {1, 5, 10, 15, 20, 25, 30, 40, 50, 60, 70, 80, 90}. This scheme is named as $P_I$ and allows
the user an advantage to select among various cases, suitable for their purpose.
In order to examine and test the $P_I$ scheme, the relative root mean square errors (RMSEs) between the MSTM
modelled and fitted values of optical properties were measured. Fig. 14 shows the values of relative RMSEs over
a range of $D_{mob}$ for the cases of  $\lambda$ (nm) = {660}; $f_{organics}$ (%) = {50}; and $D_f$ = {1.5, 1.6, 1.7, 1.8, 1.9, 2.2}. For the
entire range of $D_{mob}$ and $D_f$, the errors in cross-sections are less than 1%. The relative RMSE is < 2.5% for SSA
and up to 16% for $g$.
Similarly, relative RMSE values for the entire range of $f_{organic}$ can be seen in Fig. 15. For the results shown in
Fig, 15, the case with values of $\lambda$ (nm)= {660}; $D_f$ = {1.7}; and $f_{organics}$ (%) = {1, 5, 10, 15, 20, 25, 30, 40, 50, 60,
70, 80, 90} were used. The errors in the cross-sections are comparable to Fig. 11, being < 1.5% in all cases.
Similarly, the relative RMSE for SSA is < 3%. The error in $g$ peaks to 18% at $f_{organics}$ < 20% for larger sizes.

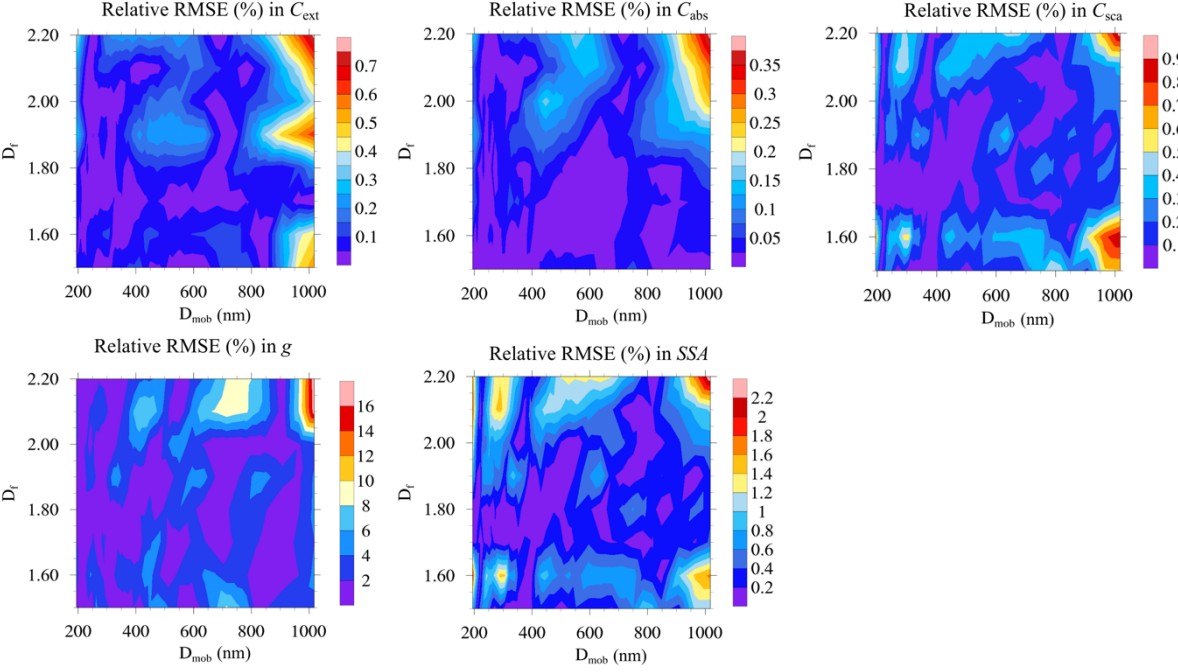

**Figure 14.** The relative RMSE between MSTM modelled and parametrized values of $C_{ext}$, $C_{abs}$, $C_{sca}$, $g$, and $SSA$
for various cases of fractal dimension ($D_f$) at $\lambda$ = 660nm. In this case, the fraction of organics ($f_{organics}$) amounts to
707 50%.


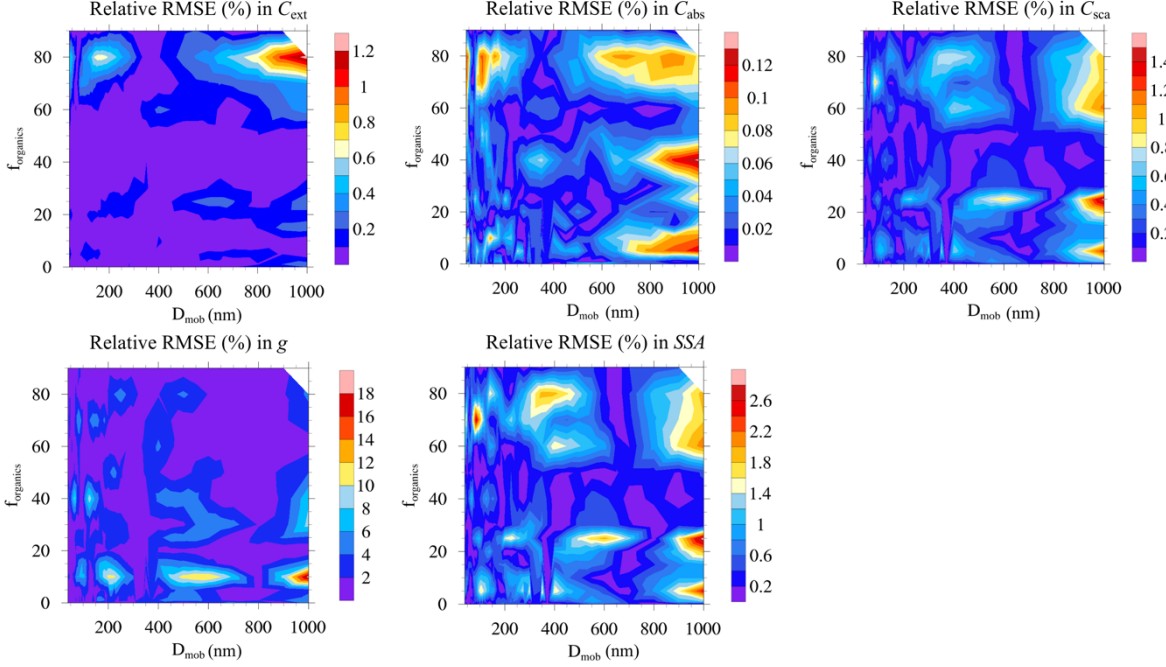

**Figure 15.** The relative RMSE between MSTM modelled and parametrized values of $C_{ext}$, $C_{abs}$, $C_{sca}$, $g$, and $SSA$
for various cases of fraction of organics ($f_{organics}$) at $\lambda = 660$nm. The fractal dimension ($D_f$) is fixed to 1.7.

It is expected, that a dataset of BCFA optical properties with higher resolution for the individual parameters
gives better results. To demonstrate this, the $P_I$ scheme is compared to another scheme $P_{II}$ with a reduced dataset.
In the $P_{II}$ scheme, the same fits were applied, but optical properties were averaged for $D_f$ in the range from 1.5 to
1.7 and $f_{organics}$ in the range from 60-90% to obtain the "averaged" fit coefficients. The errors from this
parametrization scheme $P_{II}$ were compared to the errors from their corresponding cases ($D_f = 1.7$, and $f_{organics} =$
60%) within the parametrization scheme $P_I$. The results are summarized in the Table 5. The relative RMSE errors
for $P_{II}$ are evidently larger than for $P_I$, and gives evidence that the dataset with higher resolution minimizes errors
when deriving parametrization schemes.

**Table 5.** Comparison between the Relative RMSE errors of parametrization schemes over a single case of BCFA
($D_f = 1.7$, $f_{organics} = 60\%$, and $\lambda = 660$nm). The errors on the left ($P_I$) are for the original scheme developed in this
study. Whereas the errors on right show the errors resulting from $P_{II}$, which is the condensed form of $P_I$ i.e., $D_f =$
1.5-1.7, and $f_{organics} = 60$-90%.

| Optical property | Relative RMSE (%) | |
|---|---|---|
| | $P_I$ | $P_{II}$ |
| $C_{ext}$ | 0.09 | 4.98 |
| $C_{abs}$ | 0.02 | 1.42 |
| $C_{sca}$ | 0.30 | 9.23 |
| $g$ | 1.17 | 8.46 |
| $SSA$ | 0.68 | 7.12 |

## 4    Conclusions
Optical properties of pure and coated BCFAs were systematically investigated as a function of particle size ($D_{mob}$),
primary particle size ($a_o$), morphology ($D_f$), composition ($f_{organics}$), and wavelength ($\lambda$), developing a
parametrization scheme for BCFA optical properties.
Modelled optical properties of BCFAs were found to be sensitive to changes in the radius of the primary particle
($a_o$) at a fixed $D_{mob}$. The highest sensitivity was seen for cross-sections ($C_{ext}$, $C_{abs}$, and $C_{sca}$), increasing by a factor
of almost ten when $a_o$ is changed from 15nm to 30nm, at a fixed $D_{mob}$. When the volume equivalent radius $R_{equ}$
of a BCFA is fixed, the values of $C_{ext}$ and $C_{abs}$ with changing $a_o$ were constant, also shown by the study of Kahnert,
2016b.
In addition to the dependency of BCFA cross-sections over size, a size dependency in optical parameters $SSA$,
$g$, and $MAC_{BC}$ was also seen. All the BCFA optical properties showed dependencies over morphology and
composition, the nature of this dependencies being specific to each optical property and size dependent. In terms
of morphology, the $C_{sca}$, $SSA$, and $g$ showed the highest sensitivity towards $D_f$, pronouncing as the BCFA grows
in size. In contrast to the results of $C_{sca}$, $SSA$, and asymmetry parameter, the $C_{ext}$, $C_{abs}$, and $MAC_{BC}$ were more
sensitive with respect to changing composition of BCFAs. For e.g., the values of $MAC_{BC}$ increased by a factor of
1.5 with increasing amount of $f_{organics}$ up to 90%, at $\lambda = 660nm$. The optical properties $SSA$, $g$, and $MAC_{BC}$ are
needed to simulate the BC radiative forcing in global models. Therefore, the simplified core-shell representation
of BC in global models does not adequately consider the above discussed impacts of morphology over the BC
optical properties.
In the visible range, the decrease in the optical properties $C_{ext}$, $C_{abs}$, $C_{sca}$, and $MAC_{BC}$ with $\lambda$ was large, whereas,
a smaller decrease in $SSA$, and $g$ with $\lambda$ was shown. The nature of the spectral dependencies with respect to
changing morphology and composition varied for various optical properties. While the other optical properties
had a less significant spectral dependence on morphology, the asymmetry parameter ($g$) showed the highest
sensitivity, dominant at a higher $D_f$, i.e., for compact aggregates. For e.g., the ratio of $g$ at $\lambda = 467nm$ and $\lambda =$
660nm changed from 1.1 to 2.6 when going from lower to higher values of $D_f$. All the cross-sections and black
carbon mass absorption cross-section $MAC_{BC}$ showed a significant increase in the spectral dependency with
increasing fraction of organics $f_{organics}$. For e.g., the spectral dependency of $MAC_{BC}$ increased from a factor of 1.97
at 1% fraction of organics to a factor of 4 at 90% fraction of organics. It was shown that the $MAC_{BC}$ for a BCFA
can be very high for the cases with high organic content, like 20 $m^2/g$ for 90% fraction of organics at $\lambda = 467nm$.
The dependencies of the Absorption Ångström Exponent (AAE) on morphology and composition were
investigated. The values of AAE changed from 1.06 up to 3.6 depending on the fraction of organics ($f_{organic}$), fractal
dimension ($D_f$), and size ($D_{mob}$). It is evident from the results, that the AAE of black carbon particle without
organic coating can significantly differ to values of about unity, contradicting the interpretation of AAE in some
studies. For e.g., the interpretation of the measurement values of AAE $\gg 1$ as biomass burning aerosol might be
misleading in the Sandradewi model (Sandradewi et al., 2008). The values of the absorption enhancement factor
($E_\lambda$) via coating calculated from the MSTM model varied from 1.0 to 3.0 as a function of wavelength ($\lambda$) and size
($D_{mob}$), whereas, the Mie theory derived $E_\lambda$ varied from 1.0 to 3.5. The ratio between the MSTM and Mie derived
$E_\lambda$ changed from 1.1 to 1.5 as a function wavelength ($\lambda$). The largest discrepancies between the MSTM and Mie
derived $E_\lambda$ was seen at the red wavelength ($\lambda = 660nm$) due to the presence of Mie resonances in larger particles.
The key message of this study is that the sensitivity of various optical properties, especially $SSA$, $g$, and $MAC_{BC}$
towards changing morphology and composition can be significant. Further, to understand the atmospheric and
climate implications of our findings, a sensitivity study on the black carbon radiative forcing $\Delta F_{TOA}$ was
conducted. It was shown that the black carbon radiative forcing $\Delta F_{TOA}$ ($Wm^{-2}$) can decrease up to 61% as the
BCFA becomes more compact in morphology i.e., a higher fractal dimension ($D_f$). Therefore, the influence of
morphology over the top of the atmosphere radiative forcing is neglected when using the simplified core-shell
representation of BC in global model simulations. With respect to changing composition, a decrease of more than
50% in $\Delta F_{TOA}$ was seen as the organic content of particle increases. These findings are particularly relevant for
modelling of polluted urban environments. It is generally assumed that the impact of BC particle becoming more
compact, and the increase in organic content are linked. It was shown that the changes in these two ageing factors
in tandem result in an overall decrease in the $\Delta F_{TOA}$. Therefore, these factors must be kept under consideration
when modelling absorption of BC containing particles and for assessing the radiative impacts using global models.
The parametrization scheme provides the user an option to estimate the BC optical properties (extinction cross-
section $C_{ext}$, absorption cross-section $C_{abs}$, scattering cross-section $C_{sca}$, single scattering albedo $SSA$, and
asymmetry parameter $g$) at the desired BC size for various combinations of $\lambda$, $D_f$, and $f_{organics}$. Even though simple
linear regression models were used in this study, the parametrization scheme showed low differences between the
parameterized and tabulated MSTM modelled values of optical properties. For the entire parametrization scheme,
the relative root mean square errors (RMSEs) in $C_{ext}$, $C_{abs}$, and $C_{sca}$ were less than 1%. Similarly, the relative
RMSE for SSA was less than 3%. The largest error of about 18% was found in $g$ at $f_{organics}$ less than 20% for larger
sizes. It must be noted that the proposed parametrisation scheme is able to accurately predict the BC optical
properties above $D_{mob}$ of 50nm under various scenarios not including uncertainties due to a fixed primary particle
size and refractive index.
It is acknowledged that the results from the parametrization scheme might vary to the results from laboratory
and ambient measurements. To understand the nature of discrepancy in modelled optical properties, we encourage
users to compare results of this study to results from laboratory or ambient measurements if applicable. It is
important to mention that the parametrisation schemes and databases based on realistic representation of BC, like
the one developed in this study, is a successful step forward towards a more accurate characterization of BC
containing particles and radiative forcing in climate models. Therefore, further studies should be conducted
developing more comprehensive databases that include more information on primary particle size, composition,
physical variables like hygroscopicity, and optical parameters like refractive indices.
**Acknowledgement**
This work is supported by the 16ENV02 Black Carbon project of the European Union through the European
Metrology Programme for Innovation and Research (EMPIR).

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
