# Peer review of "Optical properties of coated black carbon aggregates: numerical simulations, radiative forcing estimates, and size-resolved parametrization scheme"

_Atmospheric Chemistry and Physics, 2020_

## Author Comment (AC1)

**Reviewer #1 (R1):**

We thank the reviewer for his/her insightful review and constructive comments. We highly appreciate your time in reviewing the manuscript. All the comments/suggestions were taken into consideration and incorporated in the revised manuscript, which has improved the quality of the revised manuscript. The point by point response to all the comments and suggestions of reviewer #1 (R1) is provided in the following sections For clarity, the reviewer's comments are provided in blue, the author's response (AR) is in black, and the revised parts of the manuscript are shown in red.

**R1 General remarks:** The manuscript by Dr. Romshoo et al. reveals the influences of BC microphysical and coating properties on its optical and radiative properties. Idealized particles and MSTM are used to give optical properties of BCFAs with coating, and a simple estimation on the radiative is used for the radiative forcing. As expected, the results indicate different influences of different microphysical properties on different optical properties, and a parameterization scheme is presented to estimate BC optical properties. Overall, the manuscript is technically solid and well presented. However, most of the findings are not completely new, and the results are not well discussed. The manuscript could be considered for publication with the following concerns being addressed.

**AR:** The authors thank the reviewer for the constructive general remarks. As suggested, the novelty of the study is highlighted in a more prominent way in the revised manuscript. The discussion of results have been improved in the revised manuscript, and the details are given in the point-by-point response to the specific comments of the Reviewer below.

**Specific comments:**

**R1 C1:** The novelty of the manuscript is not well presented. As also noticed by the authors, there have been a large amount of numerical studies on the optical and radiative properties of BC with complex coating and morphology. Either the fractal aggregate model or coating scheme has been considered before, and most of the conclusions are also noticed before by similar studies. I think there are may be multiple papers with similar titles. Thus, the authors have to better demonstrate the uniqueness of this study.

**AR:** Thank you for asking to highlight the novelty of the manuscript with respect to the previous studies on the subject. Although, we have briefly mentioned the novelty of the research in the paper, however, the new findings of the study are summarized as follows.

    i.        There have been numerous modeling-based studies in the subject area of optical properties of black carbon fractal aggregates (BCFAs). However, the currently available databases/parametrisations lack information about the parameter 'external coating of the BCFAs'. The reason for this might be the computational load for such a task is substantial due to the time-consuming simulations. In this work, we covered this gap by taking up the computationally expensive task and investigate the optical properties of BCFAs under an elaborated systematic approach. In addition to the parameter of external coating, defined by fraction of organics ($f_{organics}$), the BCFAs were studied as a function of the radius of the primary particle ($a_o$), fractal dimension ($D_f$), wavelength ($\lambda$), and mobility diameter ($D_{mob}$).

    ii.      The study is the first of its kind that expresses the modelled optical properties of BC in terms of mobility diameter ($D_{mob}$) instead of the number of primary particles ($N_s$) or volume equivalent radius ($R_{vol}$) which are mostly used in modelling studies (Smith and Grainger, 2014; Kahnert, 2010a; Liu et al., 2019; Luo et al., 2018). Using the conversion model (Sorensen, 2011), the modelled optical results in terms of the mobility diameter ($D_{mob}$) are more relevant and relatable to ambient and laboratory BC studies.

    iii.    The spectral dependency of the optical properties of purely BCFAs have been studied (Kahnert, 2010a; Smith and Grainger, 2014). Previous modelling studies have calculated the optical properties of coated BCFAs at a wavelength of 550nm (Liu et al., 2017; Luo et al., 2018). In this study, in addition to discussing the spectral dependency of the optical properties of coated BCFAs, it is demonstrated how the spectral dependency is also a function of morphology and composition in the visible wavelength range.

iv.      Various parametrization schemes and databases have been developed in the previous studies over the optical properties of pure BCFAs without coating (Smith and Grainger, 2014; Kahnert, 2010a; Liu et al., 2019; Luo et al., 2018). This study investigated the effect of external coating over the BC optical properties and a parametrization scheme for optical properties of coated BCFAs is suggested. The size-resolved parametrization scheme for the optical properties of the coated BCFAs is provided for various fractal dimension ($D_f$) and fraction of organics ($f_{organics}$). The proposed parametrization scheme is applicable for modelling, ambient and laboratory-based BC studies.

v.      The relative changes in the top of the atmosphere radiative forcing were calculated as a function of fractal dimension ($D_f$) and fraction of organics ($f_{organics}$). Sensitivity analysis showed that these ageing factors in tandem could cause changes in the dynamics of the boundary layer under certain conditions.

Regarding the key points summarized above; the "Introduction" section of the manuscript is modified to highlight the novelty of this study in a pronounced manner:

Discrepancies due to Mie theory have caused an increasing interest in the simulation of the BC optical properties assuming a more realistic fractal morphology. The size-dependent empirical formula for the optical properties of BCFAs was derived for the wavelength range from 200nm up to 12.2µm (Kahnert et al., 2010). The optical properties of pure BCFAs, i.e., without any external coating, were investigated by Smith and Grainger (2014), further developing a parametrization for optical properties of pure BCFAs with respect to the number of primary particles ($N_s$). A method to estimate the optical properties BCFAs was proposed using the machine learning method, support vector machine (Luo et al., 2018). Empirical equations on the BC Ångstrom absorption exponent (AAE) were derived for different BC morphologies (Liu et al., 2018). A database containing optical data was developed that includes the aggregation structure, refractive index, and particle size of BCFAs (Liu et al., 2019).

However, the previous modelling-based studies were not able to take into account the information about the parameter: external coating of the BCFAs. The reason for this could be that the time-consuming simulations make the computational load for such a task substantially large. Additionally, various ambient and laboratory studies have emphasized the role of organic external coating in influencing the BC absorption and scattering properties (Zhang et al., 2008, Ouf et al., 2016; Dong et al., 2018, Shiraiwa et al., 2010). It was also pointed out that improved size-resolved datasets and models for the light absorbing carbon (LAC) is required that includes observables like optical properties, OC/BC ratio, burning phase or fuel types (Liu et al., 2020). Therefore, a size-resolved parametrization scheme for optical properties of BCFAs including the external coating parameter is very important.

This investigation involved computationally intensive modeling aimed at understanding and quantifying the changes that BCFAs and their optical properties undergo by simulating various cases of the BCFAs under an elaborated systematic approach that is designed to span a wide parameter space. The external coating parameter is quantified through the fraction of organics ($f_{organics}$). The BCFAs cases are classified according to various $f_{organics}$, morphologies, and wavelengths. This approach of categorization involving the $f_{organics}$ of BCFAs is aimed to bridge the gaps that are present in the modeled optical data from the previous studies. The optical properties were calculated using the T-matrix code (Mackowski et al., 2013) and the findings are presented and discussed with respect to the equivalent mobility diameter ($D_{mob}$) making it more relevant and comparable for laboratory, and ambient studies in which mobility spectrometers are often used for size classification.

The study highlights how modifications in the morphology and $f_{organics}$ of BCFAs can further influence the BC radiative forcing. Finally, the parameterization scheme for optical properties (extinction, scattering, and absorption) of coated BCFAs was developed as a function of size for different morphologies, $f_{organics}$, and wavelengths.

Liu, D., He, C., Schwarz, J. P., and Wang, X.: Lifecycle of light-absorbing carbonaceous aerosols in the atmosphere, npj Clim Atmos Sci, 3, 40, doi: 10.1038/s41612-020-00145-8, 2020.

Liu, C., Xu, X., Yin, Y., Schnaiter, M. and Yung, Y. L.: Black carbon aggregates: A database for optical properties, J. Quant. Spectrosc. Radiat. Transf., doi: 10.1016/j.jqsrt.2018.10.021, 2019.
.

**R1 C2:** The coating model considered in this study is not well described. Core-shell structure for each independent monomer is reasonable if the coating fraction was relative small, while becomes less realistic if the coating fraction is large. Would the results for f_coaitng=90% still be reliable?

**AR:** Thank you for the comment. We agree with the reviewer that when the coating fraction is large (>80%), the coating model seems unrealistic. However, we expect that the morphology of the aggregate also plays a role while applying this coating model. Therefore, keeping the above facts in mind, we have used the coating model only for BC fractal aggregates with fractal dimension below 2.2. In such cases, where the BC aggregate does not have a completely compact structure, the results are expected to be reliable (Luo et al., 2018). Moreover, it was shown by Kahnert et al., 2017 that the coating model (closed-cell model) used in this study yields good results.

   The reason for using the coating model comprising of individually coated primary particles is given in lines 162-166. A technical description of how the coating model was applied in the study is given in lines 166-182. The lines 162-184 are rewritten for better clarity as follows:

Ouf et al. (2016) conducted NEXAFS analysis on BC produced from a diffusion flame-based mini-CAST burner and found that organics (by-products of the combustion) get attached to the edge of graphite crystallites without changing the inner structure of the core. This laboratory result can be simulated for coated BC in radiative modeling studies by assuming a spherical coating around each individual primary particle of a BC aggregate (Luo et al., 2018).

   For the sake of simplicity and computational limitations, this representation of coated BC shown in Fig. 2 (bottom panel) was chosen for the entire study. In order to simulate such BC aggregates with individually coated primary particle, the inner radius of the primary particle ($a_i$) is fixed to 15 nm. Whereas the outer radius of the primary particle ($a_o$) consisting of the organics, is varied from 15.1nm to 30nm with the fraction of organics ($f_{organics}$) changing from 1% to 90% respectively. The relationship between the outer radius of the primary particle ($a_o$), the inner radius of the primary particle ($a_i$), and the fraction of organics ($f_{organics}$) is shown below:

$$\frac{4}{3}\pi a_i^3 = (1 - f_{organics})\, \frac{4}{3}\pi a_o^3 \quad . \tag{3}$$

It must be noted that when the fraction of organics ($f_{organics}$) is large (>80%), this representation of coated BC is not completely realistic. Additionally, as the morphology of the aggregate becomes more compact, and the $f_{organics}$ > 80%, using this coated BC representation for such cases results in a practically unrealistic particle (randomly immersed BC primary particles in a spherical coating structure). Therefore, both the composition and morphology of the aggregate play a role while choosing the representation for coated BC. Keeping the above facts in mind, we have limited the use of this coating model only for coated BCFAs with fractal dimension $D_f$ below 2.2. In such cases, where the BC aggregate does not have a completely compact structure, the results are expected to be reliable (Luo et al., 2018). Moreover, Kahnert et al., 2017 compared the coating model (closed-cell model) used in this study to a realistic model, which showed good comparability.

Luo, J., Zhang, Y., Zhang, Q., Wang, F., Liu, J. and Wang, J.: Sensitivity analysis of morphology on optical properties of soot aerosols, Opt. Express, doi:10.1364/oe.26.00a420, 2018.

Kahnert, M.: Optical properties of black carbon aerosols encapsulated in a shell of sulfate: comparison of the closed cell model with a coated aggregate model, Opt. Express, doi:10.1364/oe.25.024579, 2017.

**R1 C3:** The parameterization scheme of the coated BCFA is a simple fitting for the particles considered in this study, and is such parameterization general enough for others' studies? For example, there are still significant uncertainties on BC size and refractive indices, can those variations be considered similarly to those given by previous studies (https://doi.org/10.5194/acp-18-6259-2018 and https://doi.org/10.1016/j.jqsrt.2018.10.021). This is really important, and is suggested to be better discussed even if the current variability is still relatively limited.

**AR:** We thank the reviewer for the comment and the suggestions thereof. In order to minimize the uncertainties in the parametrization scheme, a large database (from model simulations) was used for the five BC optical properties (extinction cross-section $C_{ext}$, absorption cross-section $C_{abs}$, scattering cross-section $C_{sca}$, single scattering albedo $SSA$, and asymmetry parameter $g$). For every case, the optical properties are modelled by the MSTM code for BC size (mobility diameter $D_{mob}$) between 10-1000nm. The resultant parametrization scheme was developed by applying linear regression models over the MSTM modelled optical properties with respect to BC size data points ranging from 10-1000nm. Therefore, it was made sure that the resultant parametrization scheme includes all the possible information about the BC size.

Although simple linear regression models were used in this study, the low values of root mean square errors (RMSEs) between the MSTM modelled and fitted values of optical properties (Fig.13 and Fig.14) gives confidence to the robustness of the proposed parametrization scheme. Moreover, the fitted coefficients are provided for each of the 192 combinations of morphology, external coating, and wavelength, providing a wide range of options for other studies to choose from.

Unfortunately, due to the high computational time involved in numerical modelling, we were unable to generate the modelled results for multiple values of refractive indices. The developed parametrization scheme can be chosen between various options of fraction of organics ($f_{organics}$), but only for fixed values of refractive indices taken from the study of Kim et al., 2015. We would like to mention this limitation and the really important point given by the Reviewer as a possible extension in future studies.

The above points shall be summarized in the modified portion of section 3.8 in the revised manuscript as follows:

In this study, the parametrization scheme is developed for five BC optical properties (extinction cross-section $C_{ext}$, absorption cross-section $C_{abs}$, scattering cross-section $C_{sca}$, single scattering albedo $SSA$, and asymmetry parameter $g$) with respect to BC size. In total, the fit coefficients for the five BC optical properties are provided for 192 cases comprising of various combinations of wavelengths ($\lambda$), fractal dimensions ($D_f$) and fraction of organics ($f_{organics}$) seen in Fig. 1. For each case, linear regression models were applied individually over the MSTM modelled optical properties for BC size data points ranging from 10-1000nm. The fit coefficients for the five optical properties in each case are provided in a tabular form as a supplement to this work. Therefore, the resultant parametrization scheme provides the user an option to estimate the five optical properties at desired BC size for any of the 192 combinations of $\lambda$, $D_f$, and $f_{organics}$.
   It must be noted that the MSTM modelled optical properties were generated at a fixed value of refractive indices because of limited computational resources. Therefore, parametrization scheme provided in this study is not able to account for the uncertainties in refractive index.

Further, the following two paragraphs are added separately in the discussion section:

The parametrization scheme provides the user an option to estimate the BC optical properties (extinction cross-section $C_{ext}$, absorption cross-section $C_{abs}$, scattering cross-section $C_{sca}$, single scattering albedo $SSA$, and asymmetry parameter $g$) at the desired BC size for various combinations of $\lambda$, $D_f$, and $f_{organics}$. Even though simple linear regression models were used in this study, the parametrization scheme showed low values of root mean square errors (RMSEs) between the MSTM modelled and fitted values of optical properties. It must be noted that the proposed parametrisation scheme is able to accurately predict the BC optical properties under various scenarios used in the parameter database. Therefore, the uncertainties due to fixed parameters like the BC primary particle size or the refractive index are not accounted for in this study.

It is important to mention that the parametrisation schemes and databases based on realistic representation of BC like the one developed in this study is a successful step forward towards a more accurate estimation of the BC radiative forcing in climate models. Therefore, further studies must be conducted developing such databases that include more observables like varying refractive index, hygroscopicity, and light absorbing coating.

**R1 C4:** The abstract and conclusion session generally summarize the findings from the numerical simulations, and they should also briefly discuss how these conclusions serve wide range of applications related to atmospheric and climate studies.

**AR:** As suggested by the reviewer, the following changes have been made in the "Abstract" and "conclusion" sections of the revised manuscript:

Abstract: The following lines in red have been incorporated/modified in the abstract to reflect the applications of this work for atmospheric and climate studies:

   The formation of black carbon fractal aggregates (BCFAs) from combustion and subsequent aging involves several stages resulting in modifications of particle size, morphology, and composition over time. To understand and quantify how each of these modifications influences the BC radiative forcing, the optical properties of BCFAs are modelled. Owing to the high computational time involved in numerical modelling, there are some gaps in terms of data coverage and knowledge regarding how optical properties of coated BCFAs vary over the range of different factors (size, shape, and composition). This investigation bridged those gaps by following a state-of-the-art description scheme of BCFAs based on morphology, composition, and wavelength. The BCFAs optical properties were investigated as a function of the radius of the primary particle ($a_o$), fractal dimension ($D_f$), fraction of organics ($f_{organics}$), wavelength ($\lambda$), and mobility diameter ($D_{mob}$). The optical properties are calculated using the multiple sphere T-matrix (MSTM) method. For the first time, the modelled optical properties of BC are expressed in terms of mobility diameter ($D_{mob}$), making the results more relevant and relatable for ambient and laboratory

BC studies. Amongst size, morphology, and composition, all the optical properties showed the highest variability with changing size. The cross-sections varied from 0.0001 $\mu m^2$ to 0.1 $\mu m^2$ for BCFA $D_{mob}$ ranging from 24 nm to 810 nm. It has been shown that $MAC_{BC}$ and $SSA$ is sensitive to morphology especially for larger particles with $D_{mob} > 100nm$. Therefore, while using the simplified core-shell representation of BC in global models, the influence of morphology over radiative forcing estimations is neglected. The Ångstrom absorption exponent varied from 1.06 up to 3.6 and increases with the fraction of organics ($f_{organics}$). Measurement results of AAE >> 1 are often misinterpreted as biomass burning aerosol, it was observed that the AAE of purely black carbon particle can be >> 1 in certain cases. The values of the absorption enhancement factor ($E_\lambda$) were found between 1.01 and 3.28 in the visible spectrum. The $E_\lambda$ was derived from Mie calculations for coated volume equivalent spheres, and from MSTM for coated BCFAs. Mie calculated enhancement factors were found to be larger by a factor of 1.1 to 1.5 than their corresponding values calculated from the MSTM method. It is shown that radiative forcings are highly sensitive towards modifications in morphology and composition. The black carbon radiative forcing $\Delta F_{TOA}$ (Wm$^{-2}$) decreases up to 61% as the BCFA becomes more compact, indicating that the global model calculations should account for changes in morphology. A decrease of >50% in $\Delta F_{TOA}$ was observed as the organic content of the particle increase up to 90%. Sensitivity analysis showed that the changes in the ageing factors (composition and morphology) in tandem could cause changes in the dynamics of the boundary layer under certain conditions. A parametrization scheme for optical properties of BC fractal aggregates was developed, which is applicable for modelling, ambient and laboratory-based BC studies. The parameterization scheme for the cross-sections (extinction, absorption, and scattering), single scattering albedo ($SSA$), and asymmetry parameter ($g$) of pure and coated BCFAs as a function of $D_{mob}$ were derived from tabulated results of the MSTM method. Spanning over an extensive parameter space, the developed parametrization scheme showed promisingly high accuracy up to 98% for the cross-sections, 97% for single scattering albedos ($SSA$), and 82% for asymmetry parameter ($g$).

Conclusion: The following lines/paragraphs have been added/modified in the section 4 "Conclusion" of the revised manuscript to incorporate the suggestions of the reviewer:

It is observed that for BC particles with $D_{mob} > 100nm$, the $MAC_{BC}$ and $SSA$ are sensitive to morphology implying that the influence of morphology over radiative forcing estimations is neglected when the simplified core-shell representation of BC is used in global models.

The complex dependencies of the Absorption Ångstrom Exponent (AAE) on morphology was investigated. It is evident from the results, that the AAE of black carbon particle without organics can indeed be >> 1 in certain cases. Therefore, the measurement values of AAE >> 1, often interpreted as biomass burning aerosol according to a standard model used in Aethalometers (Sandradewi et al., 2008), are misleading.

The findings of this study have important implications for the atmospheric and climate studies of BC. The black carbon radiative forcing $\Delta F_{TOA}$ (Wm$^{-2}$) can decrease up to 61% as the BCFA becomes more compact in morphology i.e., a higher fractal dimension ($D_f$). Therefore, the influence of morphology over the top of the atmosphere radiative forcing is neglected while using the simplified core-shell representation of BC in global model simlations. Whereas, there is a decrease > 50% in $\Delta F_{TOA}$ as the organic content of particle decreases i.e., a higher fraction of organics ($f_{organic}$). The findings are particularly relevant for modelling of urban pollution.

It is observed that the impact of BC particle becoming more compact, and the increase in organic content go in the same direction i.e., result in a decrease in the $\Delta F_{TOA}$. Sensitivity analysis showed that these changes in tandem could cause changes in the dynamics of the boundary layer under some scenarios. Therefore, these factors must be kept under consideration while designing the BC simulations and for assessing the radiative impacts using global models.

Sandradewi, J., Prévôt, A. S. H., Szidat, S., Perron, N., Alfarra, M. R., Lanz, V. A., Weingartner, E. and Baltensperger, U. R. S.: Using aerosol light abosrption measurements for the quantitative determination of wood burning and traffic emission contribution to particulate matter, Environ. Sci. Technol., doi:10.1021/es702253m, 2008.

R1 C5: The manuscript considers both optical properties (e.g., C_ext, C_abs, SSA and so on) and the radiative effects. To avoid misunderstanding, "optical properties" instead of radiative properties are suggested.

**AR:** Thank you for the suggestion. We agree with the Reviewer. As suggested, the term 'radiative properties' is changed to 'optical properties' to avoid misunderstandings.

**R1 C6:** The radiative forcing of coated BC has been considered by Zeng et al., which also considered the hygroscopic growth of the particles (https://doi.org/10.1029/2018JD029809), and the results from this study is suggested to be compared with theirs.

**AR:** We thank the reviewer for the suggestion. In Zeng et al., 2019, the radiative forcing of organics coated BC particles is estimated as a function of increasing hygroscopicity with the help of Santa Barbara DISORT Atmospheric Radiative Transfer (SBDART) model. In their study, they have used heavily coated particles with 90 – 99% coating fraction representing extremely aged soot. The focus of the study by Zeng et al., 2019 is over aged BC and the model used for calculation of radiative forcing is different than ours. In spite of that, the results of our study follow the findings of Zeng et al., 2019, as the BC particle becomes more hydrophilic in nature (for instance, by increase of organic content), there is a decrease the top of the atmosphere TOA radiative forcing.

As suggested, the results of Zeng et al., 2109 are compared with our results in section 3.7 of the revised manuscript:

It is observed from Table 4 that the top of the atmosphere forcing $\Delta F_{TOA}$ decreases by up to 55% as the organic content of the particles increases to 90%. This result is in agreement with the findings of Zeng et al., 2019 where the increasing hygroscopicity of the BC particle results in negative top of the atmosphere forcing. However, it must be noted that in the study of Zeng et al., 2019, the focus was over aged BC particles with 90-99% coating fraction and the Santa Barbara DISORT Atmospheric Radiative Transfer (SBDART) model was used for estimating the radiative forcing.

Further, a short discussion to include the parameter of hygroscopicity in future studies is recommended in the discussion section as follows:

It is important to mention that the parametrisation schemes and databases based on realistic representation of BC like the one developed in this study is a successful step forward towards a more accurate estimation of the BC radiative forcing in climate models. Therefore, further studies must be conducted developing such databases that include more observables like varying refractive index, hygroscopicity, and light absorbing coating.

Zeng, C., Liu, C., Li, J., Zhu, B., Yin, Y. and Wang, Y.: Optical Properties and Radiative Forcing of Aged BC due to Hygroscopic Growth: Effects of the Aggregate Structure, J. Geophys. Res. Atmos., doi:10.1029/2018JD029809, 2019.

**R1 C7:** The Mie and RDG have well be tested to result in significant errors on estimation of BCFA optical properties, and, considering that the manuscript already has a large amount of results, corresponding results on RDG and Mie are not suggested to be considered in this study.

**AR:** We thank the reviewer for the suggestion. As recommended, the optical properties calculated from RDG approximation shall be removed from the revised manuscript. However, we feel that the optical properties derived from the Mie calculations should be included in the manuscript since they are representative of aged BC particle and most commonly used in BC studies. The results of Mie calculations will be used as a reference i.e., when fractal dimension is 3, a complete spherical aged BC particle.

---

## Author Comment (AC2)

**Reviewer #2 (R2):**

The authors thank the reviewer for providing constructive comments and insightful suggestions on the manuscript. We highly appreciate your time in reviewing the manuscript. The point-by-point response to all the comments and suggestions of reviewer #2 (R2) is provided in the following sections. For easy visualization, the reviewer's comments (R2 C) are provided in blue and the author's response (AR) is given in black color below the reviewer's comment. All the comments/suggestions were taken into consideration and incorporated in the revised manuscript which has improved the quality of the revised manuscript. The revised parts of the manuscript along with the references are shown in red.

**R2 General remarks:** The manuscript presents results from modelling studies of the modification of optical properties of fractal-like black carbon (BC) particles, when the radius of the primary particles, fractal dimension, fraction of organics, wavelength, and mobility diameter are varied. The study uses the multiple sphere T-matrix (MSTM) method. Based on the results of the studies the authors estimate effects of the parameter variations on the radiative forcing by the fractal-like BC aggregates and develop a parametrisation scheme for radiative properties of BC fractal aggregates, which is applicable for modelling, ambient and laboratory-based BC studies. The study concludes with a detailed analysis of uncertainties when applying the proposed parameterisations.

The topic is of relevance for the modelling of the climate impact of BC containing aerosol particles. The study claims a significant improvement of results of climate model studies compared to the usually applied core-shell model for coated spheres using Mie theory. The parameter span used in the variation studies covers a wide range of BC properties from laboratory-generated aerosol to ambient BC containing particles. The study is well designed and systematically conducted. The presentation is concise and clear, and the topic fits well into the scope of the journal. Before being acceptable for publication, two major issues need to be tackled: the increase in knowledge compared to numerous previous studies on the radiative effects of fractal-like BC agglomerates is not clearly presented, and the discussion of results requires improvement. Details are specified in the next section.

**AR:** Thank you for the useful and constructive general remarks. As suggested, the novelty of this work with reference to the previous studies on the topic is highlighted in a more prominent way in the revised manuscript. The discussion of results has been improved in the revised manuscript, and the details are given in the point-by-point response to the specific comments of the Reviewer below.

**Specific comments:**

**R2 C1:** As stated briefly in the General Remarks, a clearer presentation of the novelty of the study and the gain in knowledge is requested. There have been numerous model studies published on the optical properties of fractal-like BC particles published, and various studies are available which conduct in-depth comparisons of model analyses with observations to quantify the discrepancy between applied theories and observations. It is recommended to explain the increase in knowledge triggered by this work in the introduction section.

**AR:** Thank you for the comment. We agree with the reviewer that the novelty of this work must be highlighted with reference to the past modeling studies on the optical properties of BC fractal aggregates and their comparisons to the modeled results with other observations. Before explaining and highlighting the novel research conducted in this study, we provide the context and highlights of the most relevant previous studies on the topic:

- Kahnert et al., 2010: The size-dependent empirical formula for the optical properties of BC aggregates was derived for the wavelength range from 200nm up to 12.2µm. The empirical formula derived from the optical properties modelled using the T-matrix method, can be used as an input to a radiative transfer model.

- Smith and Grainger., 2014: The radiative properties of pure BC fractal aggregate, i.e. without any external coating, were investigated, further developing a parametrization for the optical properties of pure BC fractal aggregates with respect to the number of primary particles ($N_s$).

- Luo et al., 2018: A method to estimate the optical properties of BC aggregates was proposed using a machine learning method, support vector machine.

- Liu et al., 2018: Empirical equations on the BC Ångstrom absorption exponent (AAE) was derived for different BC morphologies.

- Liu et al., 2019: A database was developed to calculate the optical properties of BC particles using the including numerical model, the multiple-sphere T-matrix method (MSTM). The database includes the aggregation structure, refractive index, and particle size of BC particles.

- Liu et al., 2020: It was emphasised that improved size-resolved datasets and models for the light absorbing carbon (LAC) is required that include information about the optical properties, OC/BC ratio, burning phase or fuel types.

Despite the numerous modeling-based studies, the developed databases/parametrisations lack information about the external coating of the BC fractal aggregate. The reason for this might be the computational load for such a task is substantial due to the time-consuming simulations. In this work, we covered the gap of the missing parameter of external coating. The external coating parameter is quantified through the fraction of organics ($f_{organics}$). The computationally expensive task of investigating the optical properties of BC fractal aggregates as a function of the radius of the primary particle ($a_o$), fractal dimension ($D_f$), fraction of organics ($f_{organics}$), wavelength ($\lambda$), and mobility diameter ($D_{mob}$) was conducted. The size-resolved parametrization scheme for the optical properties of the coated BC fractal aggregates is provided at various fractal dimension ($D_f$) and fraction of organics ($f_{organics}$).

Accordingly, the lines 80-97 in 'Introduction' section of the preprint were rewritten in the revised manuscript as follows:

Discrepancies due to Mie theory have caused an increasing interest in the simulation of the BC optical properties assuming a more realistic fractal morphology. The size-dependent empirical formula for the optical properties of BCFAs was derived for the wavelength range from 200nm up to 12.2μm (Kahnert et al., 2010). The optical properties of pure BCFAs, i.e., without any external coating, were investigated by Smith and Grainger (2014), further developing a parametrization for optical properties of pure BCFAs with respect to the number of primary particles ($N_s$). A method to estimate the optical properties BCFAs was proposed using the machine learning method, support vector machine (Luo et al., 2018). Empirical equations on the BC Ångstrom absorption exponent (AAE) were derived for different BC morphologies (Liu et al., 2018). A database containing optical data was developed that includes the aggregation structure, refractive index, and particle size of BCFAs (Liu et al., 2019).

However, the previous modelling-based studies were not able to take into account the information about the parameter: external coating of the BCFAs. The reason for this could be that the time-consuming simulations make the computational load for such a task substantially large. Additionally, various ambient and laboratory studies have emphasized the role of organic external coating in influencing the BC absorption and scattering properties (Zhang et al., 2008, Ouf et al., 2016; Dong et al., 2018, Shiraiwa et al., 2010). It was also pointed out that improved size-resolved datasets and models for the light absorbing carbon (LAC) is required that includes observables like optical properties, OC/BC ratio, burning phase or fuel types (Liu et al., 2020). Therefore, a size-resolved parametrization scheme for optical properties of BCFAs including the external coating parameter is very important.

This investigation involved computationally intensive modeling aimed at understanding and quantifying the changes that BCFAs and their optical properties undergo by simulating various cases of the BCFAs under an elaborated systematic approach that is designed to span a wide parameter space. The external coating parameter is quantified through the fraction of organics ($f_{organics}$). The BCFAs cases are classified according to various $f_{organics}$, morphologies, and wavelengths. This approach of categorization involving the $f_{organics}$ of BCFAs is aimed to bridge the gaps that are present in the modeled optical data from the previous studies. The optical properties were calculated using the T-matrix code (Mackowski et al., 2013) and the findings are presented and discussed with respect to the equivalent mobility diameter ($D_{mob}$) making it more relevant and comparable for laboratory, and ambient studies in which mobility spectrometers are often used for size classification.

The study highlights how modifications in the morphology and $f_{organics}$ of BCFAs can further influence the BC radiative forcing. Finally, the parameterization scheme for optical properties (extinction, scattering, and absorption) of coated BCFAs was developed as a function of size for different morphologies, $f_{organics}$, and wavelengths.

Liu, D., He, C., Schwarz, J. P., and Wang, X.: Lifecycle of light-absorbing carbonaceous aerosols in the atmosphere, npj Clim Atmos Sci, 3, 40, doi: 10.1038/s41612-020-00145-8, 2020.

Liu, C., Xu, X., Yin, Y., Schnaiter, M. and Yung, Y. L.: Black carbon aggregates: A database for optical properties, J. Quant. Spectrosc. Radiat. Transf., doi: 10.1016/j.jqsrt.2018.10.021, 2019.

**R2 C2:** Introduction section and later: Some key references should be included and discussed:

**AR:** Thank you for sharing some important and useful references. All the suggested references were included in the revised manuscript, further details given below for each sub-comment in blue.

- Liu et al. (2020) discussed the life cycle of light-absorbing carbonaceous aerosols in the atmosphere, this paper should be included in the introduction section;

  **AR:** In the study by Liu et al. (2020), the requirement of incorporating observables such as OC/BC ratio, size distribution, degree of internal mixing, and hygroscopicity in the models and inventories of light absorbing carbon (LAC) has been highlighted. As suggested, the following lines about the study by Liu et al. (2020) have been added to the 'Introduction' section of the revised manuscript:

  It was also pointed out that improved size-resolved datasets and models for the light absorbing carbon (LAC) are required that includes observables like optical properties, OC/BC ratio, burning phase, or fuel types (Liu et al., 2020). Therefore, a size-resolved parametrization scheme for optical properties of BCFAs including different compositions is needed.

  Liu, D., He, C., Schwarz, J. P., and Wang, X.: Lifecycle of light-absorbing carbonaceous aerosols in the atmosphere, npj Clim Atmos Sci, 3, 40, doi: 10.1038/s41612-020-00145-8, 2020.

- one of the key papers on the optical properties of fractal-like aggregates by Berry and Percival (1986) is missing, here the authors discuss already that optical properties of fractal-like aggregates are determined by the primary spheres;

  **AR:** In the theoretical study by Berry and Percival (1986), the optical cross-sections of absorbing spheres are computed as a function of number of spheres ($N_s$), fractal dimension ($D_f$), and complex refractive index. It is shown that the optics of clusters can be very different, strongly depending on whether the fractal dimension ($D_f$) is less than 2 or greater than 2. As suggested by the Reviewer, the following changes are made in the revised manuscript:

  After line 58 in the 'Introduction' section of the preprint:

  It was theoretically shown in clusters of absorbing spherules that the change in the optical cross-sections with an increasing number of spherules (aggregation) is strongly dependent on the morphology (Berry and Percival, 1986).

  After line 393 in the 'Results and discussion' section of the preprint:

  The dependency of the optical cross-section over the fractal dimension ($D_f$) was pointed out by Berry and Percival, 1986 where the change in the cross-sections depends on whether the fractal dimension ($D_f$) is less than 2 or greater than 2.

  Berry, M. V., and Percival, I. C.: Optics of fractal clusters such as smoke, Opt. Act., 33, 577-591, doi: 10.1080/713821987, 1986.

- the entire discussion of non-fractal light absorbing carbonaceous matter in the atmosphere from biomass burning is missing (Chakrabarty et al., 2010; Chen and Bond, 2010; Chung et al., 2012; Feng et al., 2013; Fleming et al., 2020). This aerosol type plays an important role in atmospheric light absorption but is not concerned by the proposed parameterisation. It needs to be clearly expressed that the proposed parameterisations do not apply to this aerosol type which, however, contributes significantly to the light absorbing carbonaceous aerosol.

  **AR:** Recent studies have reported a class of organic carbon (OC) with light absorbing properties, known as brown carbon (BrC). The imaginary part of the BrC refractive index increases towards shorter wavelengths, strongly absorbing solar radiation in the blue and near-ultraviolet spectrum. We agree with the reviewer that this aerosol plays an essential role in atmospheric light absorption. In this study, we focused on non-coated and organic coated BC aggregates. If BrC coated BC aggregates was included, the optical properties were to be computed for an addition set of refractive indices. Unfortunately, due

to the time consuming nature of the simulations, we were not able to include BrC coatings. It will be very interesting to conduct a similar study for BC fractal aggregates with absorbing organic coating in the future.

As suggested by the Reviewer, the following changes are made in the revised manuscript:

After line 62, in the 'Introduction' section of the preprint, the concept was introduced:

Additionally, there exists a class of organic carbon (OC) with light absorbing properties, known as brown carbon, strongly absorbing solar radiation in the blue and near-ultraviolet spectrum (Fleming et al., 2020; Feng et al., 2004; Chakrabarty et al., 2010; Chen and Bond, 2010).

After line 166, in the 'Methods' section of the preprint:

It must be noted that the focus of this study is on BCFAs with coatings consisting of non-absorbing organics. If a brown carbon coating was to be included in the study, information and extra computational time regarding their refractive indices was needed. Unfortunately, due to the time-consuming nature of simulations, the generated database could not include BCFAs with brown carbon coating.

In 'Conclusion' section, a short outlook was given:

It is important to mention that the parametrisation schemes and databases based on realistic representation of BC like the one developed in this study is a successful step forward towards a more accurate estimation of the BC radiative forcing in climate models. Therefore, further studies must be conducted developing such databases that include more observables like varying refractive index, hygroscopicity, and light absorbing coating.

Chakrabarty, R. K., Moosmueller, H., Chen, L. W. A., Lewis, K., Arnott, W. P., Mazzoleni, C., Dubey, M. K., Wold, C. E., Hao, W. M., and Kreidenweis, S. M.: Brown carbon in tar balls from smoldering biomass combustion, Atmos. Chem. Phys., 10, 6363-6370, doi: 10.5194/acp-10-6363-2010

Fleming, L. T., Lin, P., Roberts, J. M., Selimovic, V., Yokelson, R., Laskin, J., Laskin, A., and Nizkorodov, S. A.: Molecular composition and photochemical lifetimes of brown carbon chromophores in biomass burning organic aerosol, Atmos. Chem. Phys., 20, 1105-1129, doi: 10.5194/acp-20-1105-2020, 2020.

Chen, Y., and Bond, T. C.: Light absorption by organic carbon from wood combustion, Atmos. Chem. Phys., 10, 1773-1787, doi: 10.5194/acp-10-1773-2010, 2010.

Feng, Y., Ramanathan, V., and Kotamarthi, V. R.: Brown carbon: a significant atmospheric absorber of solar radiation?, Atmos. Chem. Phys., 13, 8607-8621, doi: 10.5194/acp-13-8607-2013, 2013.

**R2 C3:** The entire topic of the impact of atmospheric processing on particle shape and resulting optical properties requires reconsideration. There are numerous reports on the change in aerosol morphology during atmospheric processing. Particularly, Liu et al. (2017) presented a detailed study on the effect of coating of fractal-like particles on the absorption properties. This study is not mentioned here although it is of high relevance since it illustrates the gradually decreasing impact of the fractal-line structure of the BC particle when becoming more and more coated; see Sections 3.3 and 3.4 of the manuscript. This effect should also be considered when discussing the absorption Ångström exponent and enhancement factors in Section 3.6.

**AR:** Thank you for the reference to the important study related to our work. In Liu et al., 2017, the relative performance of various methods used for simulating the scattering cross-section and enhancement factors, with respect to the mass ratio, MR (=$M_{non-BC}/M_{rBC}$) was reported. Liu et al., 2017 compared four methods of mixing BC and non-BC components: externally mixed, Maxwell–Garnett homogeneously mixed, Bruggemann homogeneously mixed, and the Rayleigh–Debye–Gans (RDG) model. It was found that when MR < 1, the simulated results are best represented as having no absorption enhancement, i.e., external mixing is considered. Whereas, when MR > 3, it is necessary to consider methods assuming optical lensing effect like the core-shell method. A generalized hybrid model was also developed for estimating enhancement factors.

The findings of the Liu et al., 2017 were related to our study in the following ways:

In 'Introduction' section of the preprint, line 74, a brief reference to the study was given as:

It was shown that the ratio of non-BC to BC components plays an important role in determining the performance of different methods used for simulating the BC optical properties (Liu et al., 2017).

In section 3.4, the following was added:

The gradually decreasing impact of the fractal morphology over the scattering results of coated BC particles was shown by Liu et al., 2017. In this study, it is seen in the case of a non-coated BC particle (Fig .6c), the $C_{sca}$ is more sensitive to the $D_f$, whereases, when the BC particles are coated (Fig. 7c, Fig 8c), the $C_{sca}$ is less sensitive towards $D_f$ and $f_{organics}$. It is observed that the $C_{sca}$ and $SSA$ (Fig. 8c, Fig. 8e) become more sensitive towards $D_f$ when the BCFA grows in size, therefore, the impact of the fractal morphology over the scattering results is also a function of particle size. Moreover, it must be noted that even though there is a decreasing impact of the fractal morphology over the scattering results, optical parameters like $C_{abs}$, $MAC_{BC}$, and $g$ showed significant variability towards change in the $f_{organics}$ (Fig 7a, 7b, 7e, and 7f).

In section 3.6, the following was added:

Liu et al., 2017 emphasized the role of mass ratio of non-BC to BC on the performance of various methods used for simulating the scattering cross-section and enhancement factors of BC particles. In this study, it is shown that the Ångstrom absorption exponent (AAE) calculated from just the MSTM method can show variability of up to a factor of 2 with an increase in the non-BC content ($f_{organics} > 90\%$). Similarly, it can be seen that the difference in the enhancement factors calculated from the core-shell theory and fractal assuming MSTM method can be up to by a factor of 1.1 to 1.5.

Liu, D. T., Whitehead, J., Alfarra, M. R., Reyes-Villegas, E., Spracklen, D. V., Reddington, C. L., Kong, S. F., Williams, P. I., Ting, Y. C., Haslett, S., Taylor, J. W., Flynn, M. J., Morgan, W. T., McFiggans, G., Coe, H., and Allan, J. D.: Black-carbon absorption enhancement in the atmosphere determined by particle mixing state, Nat. Geosci., 10, 184-U132, doi: 10.1038/ngeo2901, 2017.

**R2 C4:** In Section 2.4, the authors use the expression by Chylek and Wong (1995) for the calculation of the radiative forcing at TOA. A brief discussion on the relationship to the more widely used radiative forcing efficiency according to Haywood and Shine (1995) and Sheridan and Ogren (1999) should be added.

**AR:** Thank you for the suggestion. The previous studies have used the multiple reflection model (Haywood and Shine, 1995; Sheridan and Ogren, 1999) to calculate the top of the atmosphere TOA forcing for an optically thin partially absorbing aerosol. The expression by Chylek and Wong (1995) for the calculation of TOA forcing is a simplified version of the multiple reflection model with some implicit approximations. In our study, the simplified version was used since we highlighted the sensitivity of the TOA forcing towards morphology and composition of BC, and the aim was not to provide the exact radiative forcing estimates.

The information given above is added in section 2.4 of the revised manuscript as follows:

The model given by Chylek and Wong (1995) is for the calculation of TOA forcing is a simplified version of the multiple reflection model (Haywood and Shine, 1995; Sheridan and Ogren, 1999) with some implicit approximations. It is important to note that this is an analytical model which can be useful to understand the sensitivities of radiative forcing to various parameters (Chylek and Wong, 1995; Lesins et al., 2002). The simplified version was used in this study to highlight the sensitivity of the TOA forcing towards the morphology and composition of BC. However, the model cannot be used to replace the accurate direct radiative forcing calculations.

**Minor issues:**

1) Title: Suggested rephrasing: "Optical properties of … "; the novelty of the study should also be reflected in the title. Otherwise the publication becomes indistinguishable from other papers with almost similar titles.

**AR:** Thank you for the useful suggestion, we agree that the Title should reflect the novelty of the study.

Accordingly, the Title of the study is changed to: "Optical properties of coated black carbon aggregates: numerical simulations, radiative forcing estimates, and size-resolved parametrization scheme"

2) Line 55: Lab slang should be avoided and hence rephrasing of "by-products of burning like organics" to, e.g., "by-products of combustion like organic vapours" is suggested.

    **AR:** Thank you for the suggestion. The phrase "by-products of burning like organics" has been changed to "by-products of combustion like organic vapours" in the revised manuscript.

3) Line 57: Here the "reshaping of the BC particles into more spherical structures" by vapour deposition is mentioned but the fractal-like structure of fresh combustion aerosol has not been introduced before.

    **AR:** Thank you for highlighting this point. The concept early stage fractal morphology of BC has been introduced in the 'Introduction' section of the revised manuscript as:

    In the early stages of formation, BC particles consist of loosely bound agglomerates made of numerous small spherules, which collide to form strongly bound chain-like aggregates (Michelsen et al., 2017).

    Michelsen, H. A. Probing Soot Formation, Chemical and Physical Evolution, and Oxidation: A Review of In Situ Diagnostic Techniques and Needs. Proc. Combust. Inst. 2017, 36, 717−735.

4) Line 61: Do the authors mean "is less absorbing by nature"?

    **AR:** Thank you for the correction. The change has been made in the revised manuscript.

5) Line 96: Suggested rephrasing: "as a function of size for various morphologies".

    **AR:** Thank you for the suggestion. The phrase "as a function of size at various morphologies" has been changed to "as a function of size for various morphologies" in the revised manuscript .

6) Line 137: The effect that "light absorption measurements are insensitive to the radii of the primary particles" is already explained by Berry and Percival (1986); this should be mentioned here.

    **AR:** Thanks for the important reference. The following addition/change has been made in the revised manuscript:

    Contrarily, Berry and Percival (1986) showed that light absorption measurements are insensitive to the radii of the primary particles. Additionally, Kahnert (2012b) pointed out that insensitivity is present when the radii of the primary particle fall in the range of 10 – 25nm.

    Berry, M. V., and Percival, I. C.: Optics of fractal clusters such as smoke, Opt. Act., 33, 577-591, doi: 10.1080/713821987, 1986.

7) Line 218: The correct reference is Kim et al. (2015), this should be corrected throughout the manuscript.

    **AR:** Thank you for the correction. The change has been made throughout the revised manuscript.